# Membrane potential regulates Hedgehog signalling in the *Drosophila* wing imaginal disc

Maya Emmons-Bell & Iswar K Hariharan* 

## Abstract

While the membrane potential of cells has been shown to be patterned in some tissues, specific roles for membrane potential in regulating signalling pathways that function during development are still being established. In the *Drosophila* wing imaginal disc, Hedgehog (Hh) from posterior cells activates a signalling pathway in anterior cells near the boundary which is necessary for boundary maintenance. Here, we show that membrane potential is patterned in the wing disc. Anterior cells near the boundary, where Hh signalling is most active, are more depolarized than posterior cells across the boundary. Elevated expression of the ENaC channel Ripped Pocket (Rpk), observed in these anterior cells, requires Hh. Antagonizing Rpk reduces depolarization and Hh signal transduction. Using genetic and optogenetic manipulations, in both the wing disc and the salivary gland, we show that membrane depolarization promotes membrane localization of Smoothened and augments Hh signalling, independently of Patched. Thus, membrane depolarization and Hh-dependent signalling mutually reinforce each other in cells immediately anterior to the compartment boundary.

**Keywords** *Drosophila*; Hedgehog; membrane potential; signaling
**Subject Categories** Development; Membranes & Trafficking; Signal Transduction

## Introduction

During the development of multicellular organisms, cell–cell interactions have an important role in regulating cell proliferation and cell fate specification. In both invertebrates and vertebrates, the Hedgehog (Hh) signalling pathway has been implicated in patterning a number of tissues during development (Lee *et al*, 2016). Alterations of this pathway have been implicated in human diseases. Reduced Hh signalling can result in congenital abnormalities such as holoprosencephaly (Xavier *et al*, 2016), and increased activity of the pathway has been implicated in multiple types of cancer (Wu *et al*, 2017; Raleigh & Reiter, 2019). In recent years, this pathway has

been targeted pharmacologically with the goal of reducing its activity in cancers where the pathway is excessively active. Identifying all the ways that this conserved pathway is regulated is of importance in understanding its role in regulating development and for devising ways to alter its activity in disease states.

Initially discovered for its role in regulating segment polarity in *Drosophila* (reviewed in Ingham, 2018), Hh signalling has since been implicated in a multitude of developmental processes. Among the best characterized is the signalling between two populations of cells that make up the *Drosophila* wing imaginal disc, the larval primordium of the adult wing and thorax. The wing disc consists of two compartments of lineage-restricted cells separated by a smooth boundary. Posterior (P) cells make the morphogen Hedgehog, which binds to its receptor Patched (Ptc), which is expressed exclusively in anterior (A) cells. Hh has a relatively short range either because of its limited diffusion (Tabata & Kornberg, 1994; Strigini & Cohen, 1997), or because it is taken up by nearby target cells via filopodia-like protrusions known as cytonemes (González-Méndez *et al*, 2017). Hh alleviates the repressive effect of Ptc on the seven-transmembrane protein Smoothened (Smo) in A cells near the boundary, initiating a signalling cascade that culminates in the stabilization of the activator form of the transcription factor Cubitus interruptus (Ci), and expression of target genes such as the long-range morphogen Dpp (Jiang & Hui, 2008; Lee *et al*, 2016; Petrov *et al*, 2017). In turn, Dpp regulates imaginal disc patterning and growth in both compartments (Hamaratoglu *et al*, 2014).

While the role of cell–cell interactions, diffusible morphogens and even mechanical forces have been studied in regulating the growth and patterning of the wing disc, relatively little attention has been paid to another cellular parameter, membrane potential or $V_{mem}$. $V_{mem}$ is determined by the relative concentrations of different species of ions across the cell membrane, as well as the permeability of the membrane to each of these ions. These parameters are influenced by the abundance and permeability of ion channels, the activity of pumps, and gap junctions. While changes in $V_{mem}$ have been studied most extensively in excitable cells, there is increasing evidence that the $V_{mem}$ of all cells, including epithelial cells, can vary depending on cell-cycle status and differentiation status (Blackiston *et al*, 2009; Sundelacruz *et al*, 2009). Mutations in genes encoding ion channels in humans ("channelopathies") can result in congenital malformations (Plaster *et al*, 2001). Similarly, experimental manipulation of ion channel permeability can cause

Department of Molecular and Cell Biology, University of California, Berkeley, Berkeley, CA, USA
*Corresponding author. Tel: +1 510 643 7438; E-mail: ikh@berkeley.edu

developmental abnormalities in mice as well as in *Drosophila* (Dahal *et al*, 2017; Belus *et al*, 2018). Only more recently has evidence emerged that $V_{mem}$ can be patterned during normal development. Using fluorescent reporters of membrane potential, it has been shown that specific cells during *Xenopus* gastrulation and *Drosophila* oogenesis appear more depolarized than neighbouring cells (Krüger & Bohrmann, 2015; Pai *et al*, 2015). A recent study established that cells in the vertebrate limb mesenchyme become more depolarized as they differentiate into chondrocytes, and that this depolarization is essential for the expression of genes necessary for chondrocyte fate (Atsuta *et al*, 2019). However, in many of these cases, the relationship between changes in $V_{mem}$ and specific pathways that regulate developmental patterning have not been established.

Here we investigate the patterning of $V_{mem}$ during wing disc development and show that the regulation of $V_{mem}$ has an important role in regulating Hh signalling. We show that the cells immediately anterior to the compartment boundary, a zone of active Hh signalling, are more depolarized than surrounding cells, and that Hh signalling and depolarized $V_{mem}$ mutually reinforce each other. This results in an abrupt change in $V_{mem}$ at the compartment boundary.

## Results

We began by asking whether or not $V_{mem}$ is patterned in the wing imaginal disc. Wing imaginal discs from third-instar larvae were dissected and incubated in Schneider's medium containing the $V_{mem}$ reporting dye DiBAC$_4$(3) (hereafter DiBAC) at a concentration of 1.9 μM for 10 min. DiBAC is an anionic, membrane-permeable, fluorescent molecule that accumulates preferentially in cells which are relatively depolarized compared to surrounding cells due to its negative charge and has been used to investigate patterns of endogenous $V_{mem}$ in non-excitable cells in a variety of organisms (Adams & Levin, 2012; Krüger & Bohrmann, 2015; Atsuta *et al*, 2019). In contrast to patch-clamp electrophysiology, utilizing DiBAC allowed us to make comparisons of membrane potential across a field of thousands of cells.

A stripe of cells running through the middle of the pouch of the wing disc appeared more fluorescent, thus indicating increased DiBAC uptake (Fig 1A and B), suggesting that these cells are more depolarized than surrounding cells. This pattern of fluorescence was

observed in more than 35 individual wing imaginal discs and was not observed when imaginal discs were cultured in the voltage-insensitive membrane dye FM4-64 (Fig 1C–E). Patterned DiBAC fluorescence was observed at both apical (Fig 1A–A′) and basal (Fig 1B–B′) focal planes. Addition of the Na$^+$/K$^+$ ATPase inhibitor ouabain to the cultured discs, which depolarizes cells by collapsing transmembrane Na$^+$ and K$^+$ gradients, resulted in increased, and more uniform, DiBAC fluorescence (Fig 1F and G), indicating that patterned DiBAC fluorescence in wing disc tissue was contingent upon mechanisms that normally maintain $V_{mem}$.

The position of the stripe of altered $V_{mem}$ is reminiscent of the anteroposterior (A-P) compartment boundary in the wing disc, which separates two lineage-restricted populations—the A cells and the P cells. In order to identify the population of depolarized cells with respect to the compartment boundary, we examined discs expressing *UAS-RFP* under the control of *patched-Gal4* (*ptc*>RFP) that had been incubated in DiBAC. *ptc*>RFP is expressed in those cells that express the highest levels of endogenous *ptc*, which are immediately anterior to the compartment boundary. The cells in which DiBAC accumulated at higher levels correlated with expression of RFP throughout the third larval instar (Fig 1H–H″), indicating that cells anterior to the compartment boundary are more depolarized than cells across the compartment boundary in the posterior compartment. *ptc*-expressing cells in the hinge also were more fluorescent upon DiBAC staining (Fig 1H, yellow arrowheads), but for the purposes of this work we focused on the wing pouch (Fig 1H, white arrowhead). As with *ptc-Gal4* expression, the domain of relative depolarization was broader in early third-instar wing discs, becoming more and more restricted to cells just anterior to the compartment boundary over the course of developmental time (Fig 1I–I′).

### Expression and function of endogenous ion channels anterior to the compartment boundary

The resting potential, $V_{mem}$, results from the activity of a large number of different transporters of charged molecules, as well as the permeability of the membrane to each of those molecules. Thus, the relative depolarization of the region immediately anterior to the compartment boundary is unlikely to result simply from a change in the activity of a single pump or channel. However, by identifying transporters expressed in this region, it should be possible to

**Figure 1. Membrane potential is patterned in the third-instar wing imaginal disc.**

A, B   Live third-instar discs incubated in DiBAC. DiBAC fluorescence is observed in the wing imaginal disc in both apical (A–A′) and basal (B–B′) optical sections. Increased fluorescence is observed in the centre of the pouch (white arrow, B′) and at the dorsoventral (D-V) compartment boundary in the anterior compartment (yellow arrow, (B′), and inset, (D)).

C–E   Comparison of DiBAC with the voltage-insensitive dye FM4-64. Incubation of live discs in FM4-64 (E) shows more uniform fluorescence when compared to DiBAC (D). Quantitative comparison of fluorescence in the two white boxes in each panel is shown in (C). $N = 7$ discs for each treatment, data compared using an unpaired *t*-test, ** indicates $P < 0.001$, error bars are standard deviations. The region boxed in yellow in (D) is shown at higher magnification to show DiBAC fluorescence at the D-V compartment boundary.

F, G   Incubation in ouabain results in brighter and more uniform DiBAC fluorescence.

H–H″   Live wing discs expressing *UAS-RFP* anterior to the compartment boundary, under the control of *ptc-Gal4*, were incubated in DiBAC, showing that the stripe of increased DiBAC fluorescence coincides with the posterior edge of *ptc* expression. White arrowhead in (H) indicates the stripe in the wing pouch; yellow arrowheads indicate the stripe in the dorsal and ventral hinge.

I, I′   Early L3 discs incubated in DiBAC. Patterned depolarization is evident throughout the third larval instar, with the stripe of increased fluorescence becoming narrower in more mature discs with developmental time (Compare I with A, bracket in I indicates width of increased DiBAC fluorescence).

Data information: Scale bars are 100 μm in all panels, except for (A′ and B′), where scale bars are 50 μm.

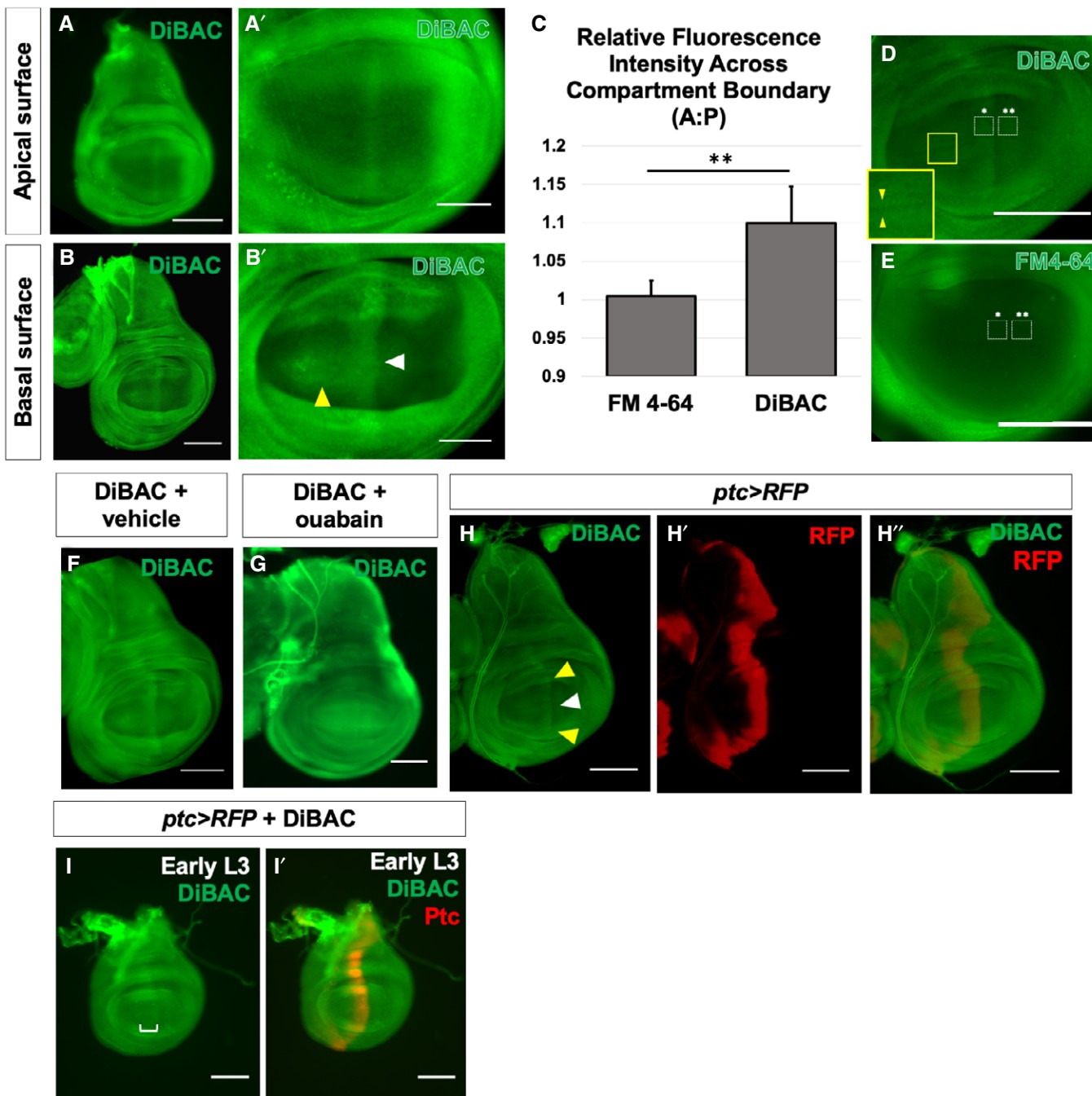

**Figure 1.**

manipulate $V_{mem}$ by altering their expression or properties. To that end, we examined a published transcriptome data set (Willsey *et al*, 2016), comparing the abundance of transcripts in *ptc*-expressing cells with those of cells in the posterior compartment. In this data set, we noticed several ion channels with differential expression between the two populations of cells. Among these are two members of the Degenerin Epithelial Na$^+$ Channel (DEG/ENaC) family of channels, *ripped pocket* (*rpk*) (Adams *et al*, 1998), and *pickpocket 29* (*ppk 29*) (Thistle *et al*, 2012). DEG/ENaC channels are members of a diverse family of amiloride-sensitive cation channels.

An antibody that recognizes the Rpk protein has been characterized previously (Hunter *et al*, 2014), which allowed us to examine its pattern of expression. In late L3 wing discs, we found that Rpk was indeed expressed anterior to the compartment boundary, in a stripe of cells that also express *ptc>RFP* (Fig 2A–A′). In addition, we observed expression of Rpk in cells near the dorsoventral (D-V) boundary. Expression was most obvious in two rows of cells flanking the D-V boundary in the anterior compartment, which are likely to be the two rows of cells arrested in the G2 phase of the cell-cycle (Johnston & Edgar, 1998). Indeed, the pattern of DiBAC

uptake in this portion of the wing disc also suggests a very thin stripe of low-fluorescence flanked by two regions of higher fluorescence (see inset in Fig 1D). This is consistent with previous work showing that cells in culture become more depolarized as they progress through S-phase and peaks at the onset of mitosis (Cone, 1969), reviewed in (Blackiston *et al*, 2009). Thus, at least in principle, the increased expression of Rpk could contribute to the depolarization observed in these regions of the wing disc.

Since Rpk, and possibly Ppk29, are expressed anterior to the compartment boundary, we tested the effect of blocking these channels by treating discs with amiloride. Amiloride is predicted to reduce the permeability of DEG/ENaC family channels (Garty, 1994). Addition of amiloride did not alter the pattern of Rpk protein expression (Fig 2C–C′). However, amiloride addition abolished the stripe of increased DiBAC fluorescence anterior to the compartment boundary (Fig 2B–B″). These findings suggest that a conductance mediated by one or more channels of the DEG/ENaC family contributes to the relative depolarization of this region. Amiloride addition did not, however, seem to affect the increased DiBAC fluorescence observed at the D-V boundary (Fig 2B″). Since amiloride likely targets multiple ENaC channels, we also depleted Rpk using an RNAi transgene expressed in cells anterior to the compartment boundary using *dpp-Gal4* and *Gal80^{TS}*, allowing us to express RNAi in only the 48 h prior to dissection. In these discs, we no longer observed the stripe of increased DiBAC fluorescence that ran along the A-P compartment boundary (Fig 2D and E white arrowhead), while increased fluorescence at the D/V boundary was preserved (Fig 2E, yellow arrowhead). Additionally, we knocked down *rpk* in the wing pouch during the last 48h of larval development using *rotund-Gal4* and *Gal80^{TS}*. *rpk* knockdown reduced anti-Rpk antibody staining (Fig EV1B and D), validating the efficacy of the dsRNA, and it also reduced DiBAC staining in the wing pouch (Fig EV1A and C). From these experiments, we conclude that reducing either the expression or permeability of endogenous DEG/ENaC family channels, notably Rpk, can abolish the region of depolarization anterior to the A-P compartment boundary. Hence, Rpk, and possibly other DEG/ENaC channels, contribute to this local alteration in $V_{mem}$.

In most cells, the Na$^+$/K$^+$ ATPase is primarily responsible for setting a negative $V_{mem}$, since it uses ATP hydrolysis to extrude three Na$^+$ ions and bring in two K$^+$ ions per cycle of activity (Morth *et al*, 2007). RNA of *ATPα* (Lebovitz *et al*, 1989), which encodes a subunit of the Na$^+$/K$^+$ ATPase, was also detected at higher levels in *ptc*-expressing cells (Willsey *et al*, 2016). Using an antibody to ATPα

(Roy *et al*, 2013), we once again observed elevated expression anterior to the compartment boundary, with a hint of increased expression at the D-V boundary (Fig 3A‴). While it is difficult to predict the contribution of patterned expression of each channel to the patterning of $V_{mem}$, detection of these channels at the anteroposterior compartment boundary of the wing disc allowed us to manipulate their expression or to pharmacologically alter their properties.

## Patterned expression of *rpk* and *ATPα* is regulated by Hedgehog signalling

Since the increased expression of Rpk and ATPα anterior to the compartment boundary occurs precisely within the region of increased Hh signalling (Fig 3A–A‴), we tested whether manipulating components of Hh signalling pathway could alter expression of Rpk or ATPα. We used a temperature-sensitive *hh* allele (*hh^{TS2}*) (Ma *et al*, 1993) in order to decrease Hh signalling for a short period of time. Larvae were raised at a permissive temperature (18°C) to permit normal *hh* function during early development and then shifted to a restrictive temperature (30°C) during the third larval instar in order to reduce *hh* function and dissected 12 h after the temperature shift. Under these conditions, increased expression of either ATPα or Rpk was not observed anterior to the compartment boundary (Fig 3B–C′, controls in Fig EV2A–A″), indicating that a normal level of *hh* activity is necessary for the increased expression of these proteins anterior to the compartment boundary. Interestingly, some expression of Rpk is still visible near the D-V boundary in the anterior compartment which is likely *hh*-independent. We then examined the effects of increasing Hh signalling. Since Hh-dependent gene expression in this region typically results from stabilization of the activator form of Cubitus interruptus (Ci) (Aza-Blanc *et al*, 1997), we generated clones of cells expressing a constitutively active version of Ci (*ci3m* (Price & Kalderon, 1999)). The *ci3m* allele has S to A mutations at PKA-phosphorylation sites 1–3, rendering the protein more resistant to proteolytic cleavage. These clones had modest increases in expression of both of Rpk and ATPα (Fig 3D–E‴). Additionally, these clones showed increased DiBAC fluorescence (Fig 3F–F′), as did clones expressing RNAi against *ptc* (Fig 3G–G′). *ci3m*-expressing clones showed increased DiBAC fluorescence in both the anterior and posterior compartments (Fig EV3A and A′), consistent with the ability of constitutively active Ci to activate Hh target genes in both compartments. Clones expressing *ptc*-RNAi only showed increased DiBAC fluorescence in the anterior compartment, as *ptc* is not expressed in the

---

**Figure 2. Expression of endogenous channels is patterned and contributes to depolarization anterior to the compartment boundary.**

A–A″  Expression of Rpk is increased anterior to the A-P compartment boundary and in two rows of cells flanking the D-V compartment boundary in the A compartment.

B–B″  Blockade of DEG/ENaC channels by incubation in amiloride abolishes the stripe of increased DiBAC fluorescence along the A-P compartment boundary (white arrowhead). White boxes indicate regions used to calculate the average DiBAC fluorescence intensity ratio across the compartment boundary = 0.98, standard deviation = 0.05, *n* = 5 discs. Increased fluorescence at the D-V boundary in the A compartment is still observable (yellow arrowhead). The same vehicle was used as in experiments with ouabain, and the control is in Fig 1F.

C, C′  Amiloride incubation does not diminish Rpk expression. White arrowhead indicates approximate position of the A-P compartment boundary, yellow arrowhead indicates approximate position of the D-V compartment boundary.

D, E  Expression of *rpk*-RNAi using *dpp-Gal4* results in diminished DiBAC fluorescence along the A-P compartment boundary (white arrowhead), while increased fluorescence at the D-V boundary in the A compartment is still observable (yellow arrowhead).

Data information: Scale bars are 100 μm, except in (B″), where scale bar is 50 μm.

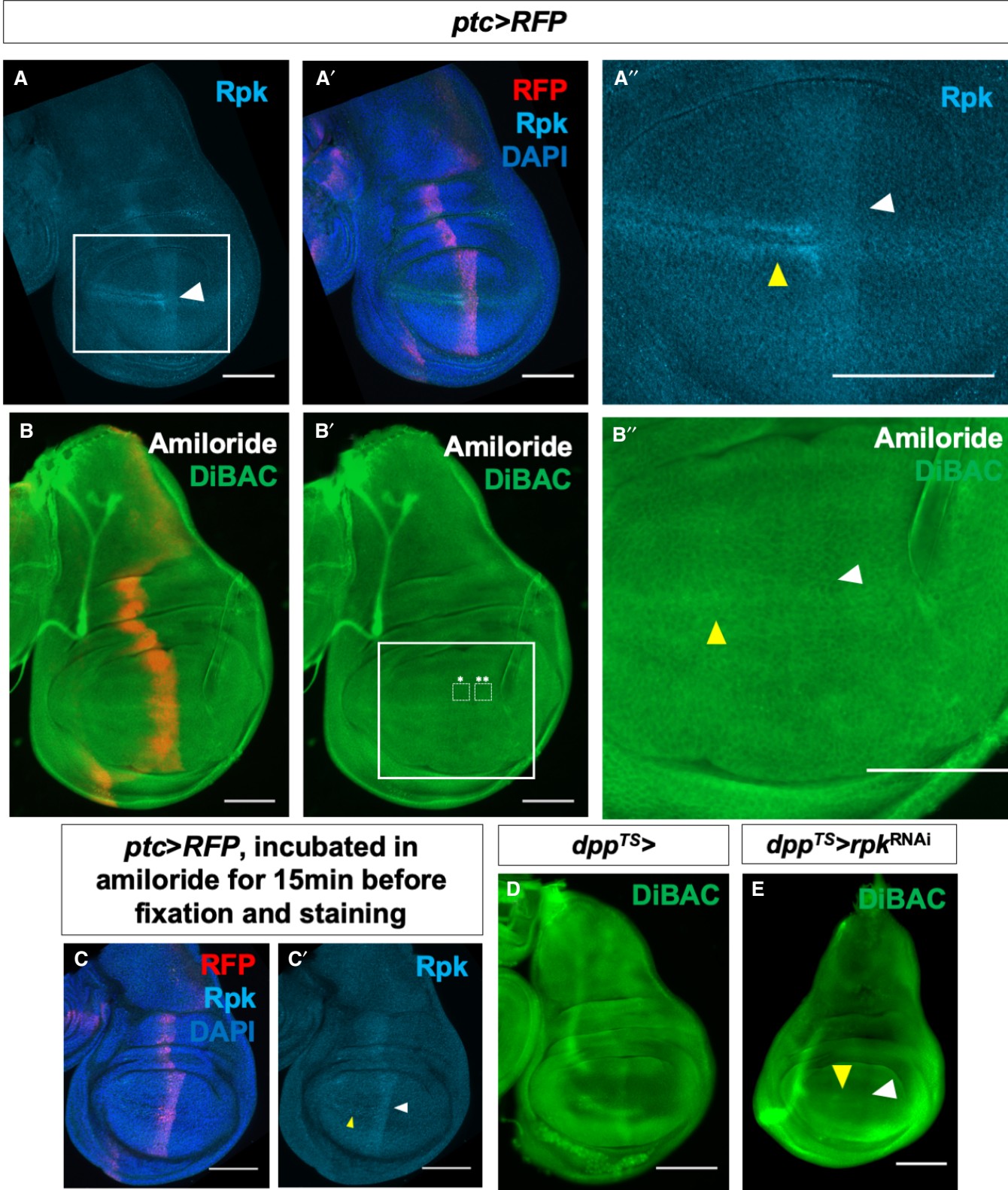

**Figure 2.**

posterior compartment (Fig EV3B and B′). There was not always perfect concordance of increased DiBAC staining with clones, so to describe this variability we quantified the ratio of DiBAC staining in *ci3m*-expressing clones to DiBAC staining to control tissue (Fig EV3C). Thus, Hh signalling appears to promote expression of both Rpk and ATPα, as well as relative depolarization of $V_{mem}$.

## Manipulating expression of endogenous ion channels modulates Hedgehog signalling

We then investigated whether a depolarized $V_{mem}$ is required for the high levels of Hh signal transduction that occur immediately anterior to the compartment boundary. To that end, we reduced *rpk* expression in the dorsal compartment of the wing imaginal disc using *ap-Gal4* (Fig EV4A) and *UAS-rpk^RNAi*. The Hh signal transducer *smo* is transcribed throughout the *Drosophila* wing imaginal disc, but immunostaining for Smo protein reveals that it is most abundant on cell membranes in the posterior compartment, and in cells directly anterior to the compartment boundary (Fig 4A–A‴). Membrane localization of Smo in the anterior compartment is thought to depend on high levels of Hh signalling (Zhu *et al*, 2003). The Hh signalling pathway is active in several rows of cells immediately anterior to the compartment boundary that receive Hh. Additionally, several components of the Hh signalling pathway are active in the entire posterior compartment, possibly because the absence of Ptc renders Smo constitutively active (Ramírez-Weber *et al*, 2000). However, target genes are not activated in posterior cells because Ci is only expressed in the anterior compartment. Knockdown of *rpk* resulted in a reduction in membrane staining of Smo in the dorsal compartment, as compared to ventral cells (Fig 4B′). This was observed in both posterior cells which do not express *ptc* and in anterior cells which do. Thus, at least in posterior cells, the effect on Smo localization does not require *ptc* function. In anterior cells near the compartment boundary, Hh signalling also results in stabilization of the activator form of Ci. In *ap>rpk^RNAi* discs, the level of activated Ci in the dorsal compartment was reduced (Fig 4B″). We also examined the expression of *ptc*, which is a direct transcriptional target of Ci (Alexandre *et al*, 1996). The stripe of staining with anti-Ptc was much fainter in the dorsal part of the disc (Fig 4C). Thus, expression of *rpk^RNAi* reduces Hh signalling in cells anterior to the compartment boundary. Knockdown of ATPα using *ap>ATPα^RNAi* resulted in severely altered tissue morphology (Fig EV4C and D), and a reduction in anti-Ptc staining (Fig EV4D), suggesting that expression of *ATPα^RNAi* also reduces Hh signalling. We have shown earlier that antagonizing the function of the Na/K-ATPase, of which

ATPα is a component, with ouabain completely abolishes the patterned depolarization in this disc (Fig 1G).

To examine the effect of altering *rpk* expression on pathways other than Hh, we reduced *rpk* expression in the wing pouch using *rotund-Gal4* and *Gal80^TS* (hereafter *rn^TS*) (Fig EV4B). Larvae were raised at 18°C and then shifted to 30°C for 48 h before dissection and staining. In these discs, anti-Smo and anti-Ci staining were clearly reduced in the wing pouch (Fig EV4E–F″). Anti-Ptc staining was also clearly reduced (Fig EV4G–H′). All of these observations were consistent with reduced Hh signalling. We also observed a modest reduction in the width of the dorsoventral stripe of Wingless expression in discs expressing *rpk*-RNAi (Fig EV4I–J″). More obvious was the absence of expression of the Notch and Wingless target gene, *cut* along the dorsoventral margin (Fig EV4K–L′). Thus, in addition to Hh signalling, altering *rpk* impacts other pathways as well.

## Manipulating $V_{mem}$ regulates Smoothened localization in salivary glands

Cells of the wing imaginal disc are quite small and columnar, posing a challenge for subcellular imaging. Cells of the salivary gland are much larger than those of the wing disc, allowing for easier visualization of subcellular protein localization. Additionally, much work characterizing the regulation of Smo has been carried out in this tissue (Zhu *et al*, 2003). To directly test whether or not altering $V_{mem}$ can modulate Hh signalling, we used the bacterial sodium channel NaChBac, which can be used to cause membrane depolarization in insect cells by overexpression (Ren *et al*, 2001; Luan *et al*, 2006; Nitabach *et al*, 2006). Expression of NaChBac in the salivary gland using the Gal4 driver line *71B-Gal4* showed a clear increase in membrane-associated staining with anti-Smo (Fig 5A, B and D) as well as increased expression of Ptc (Fig 5E and F). Correspondingly, increased expression of the mammalian potassium channel Kir2.1, which would be predicted to hyperpolarize *Drosophila* cells (Baines *et al*, 2001; Hodge, 2009), reduces fluorescence at the cell surface and increases intracellular fluorescence (Fig 5C and D). Thus, sustained alteration in $V_{mem}$ modulates Hh signalling as assessed by Smo localization and Ptc expression.

In order to investigate the short-term consequences of altering $V_{mem}$, we used channelrhodopsin ChR2, which when exposed to blue light causes membrane depolarization (Nagel *et al*, 2003; Schroll *et al*, 2006). In addition to providing a second independent way of depolarizing cells, this approach allowed us to examine short-term changes in Hh signalling that occur in response to depolarization. Dissected, ChR2-expressing salivary glands were either

---

**Figure 3.  Patterned Rpk and ATPα expression and membrane depolarization require Hh signalling.**

A–A‴  Immunostaining of discs expressing *ptc>RFP* with antibodies to Rpk and ATPα showing elevated levels of both proteins anterior to the compartment boundary. White arrowheads indicate approximate position of the A-P compartment boundary.

B–C′  L3 discs that are temperature-sensitive for *hh* following shift to the restrictive temperature for 12 h show loss of increased expression of Rpk (B, B′) and ATPα (C, C′) anterior to the compartment boundary. Controls shown in Fig EV2. White arrowheads indicate approximate position of the A-P compartment boundary.

D–E‴  Clones of cells expressing an activated form of the transcription factor Ci show elevated Rpk and ATPα expression in the anterior compartment (white box) as well as the posterior compartment (yellow arrowhead). (E–E‴) A single clone in the A compartment is shown at higher magnification.

F–F′  A clone of cells in the anterior compartment expressing *ci3M* following incubation in DiBAC.

G–G′  A clone of cells in the anterior compartment expressing *ptc*-RNAi following incubation in DiBAC. The clones also express RFP.

Data information: All scale bars are 100 μm, except for (B′), (C′) and (F–G′) where scale bars are 50 μm and (E–E‴) where scale bars are 25 μm.

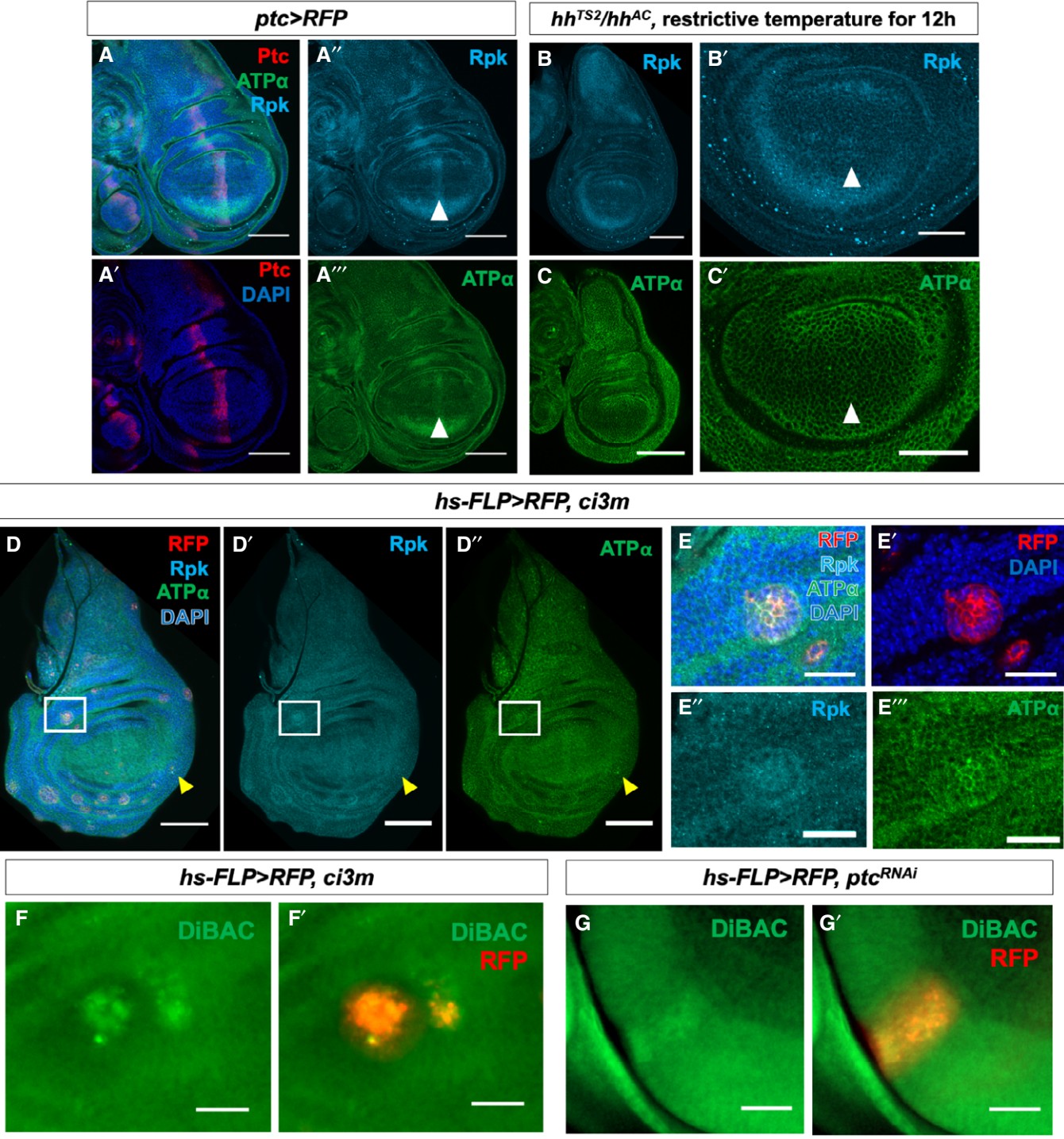

**Figure 3.**

kept in darkness or exposed to activating light for variable intervals of time. Compared to glands kept in the dark, light-exposed glands had higher levels of Smo at the cell membrane (Fig 5G–I), suggesting that even relatively short-term depolarization can facilitate Smo accumulation at the cell surface. A time course showed that an increase in Smo membrane abundance was detectable as early as 10 min and appeared maximal after 25 min of activating light

(Fig 5M–Q). A red light-activated depolarizing channelrhodopsin ReaChR (Lin *et al*, 2013; Inagaki *et al*, 2014) also elicited a similar effect (Fig EV5A–C). Activation of ChR2 did not alter membrane abundance of the integrin component Mys (Bunch *et al*, 1992) or the integrin-associated protein Talin (Brown *et al*, 2002; Fig EV5D–G′). In contrast, expression and activation of ChloC, a blue light-activated anion channel which hyperpolarizes cells (Wietek *et al*, 2014;

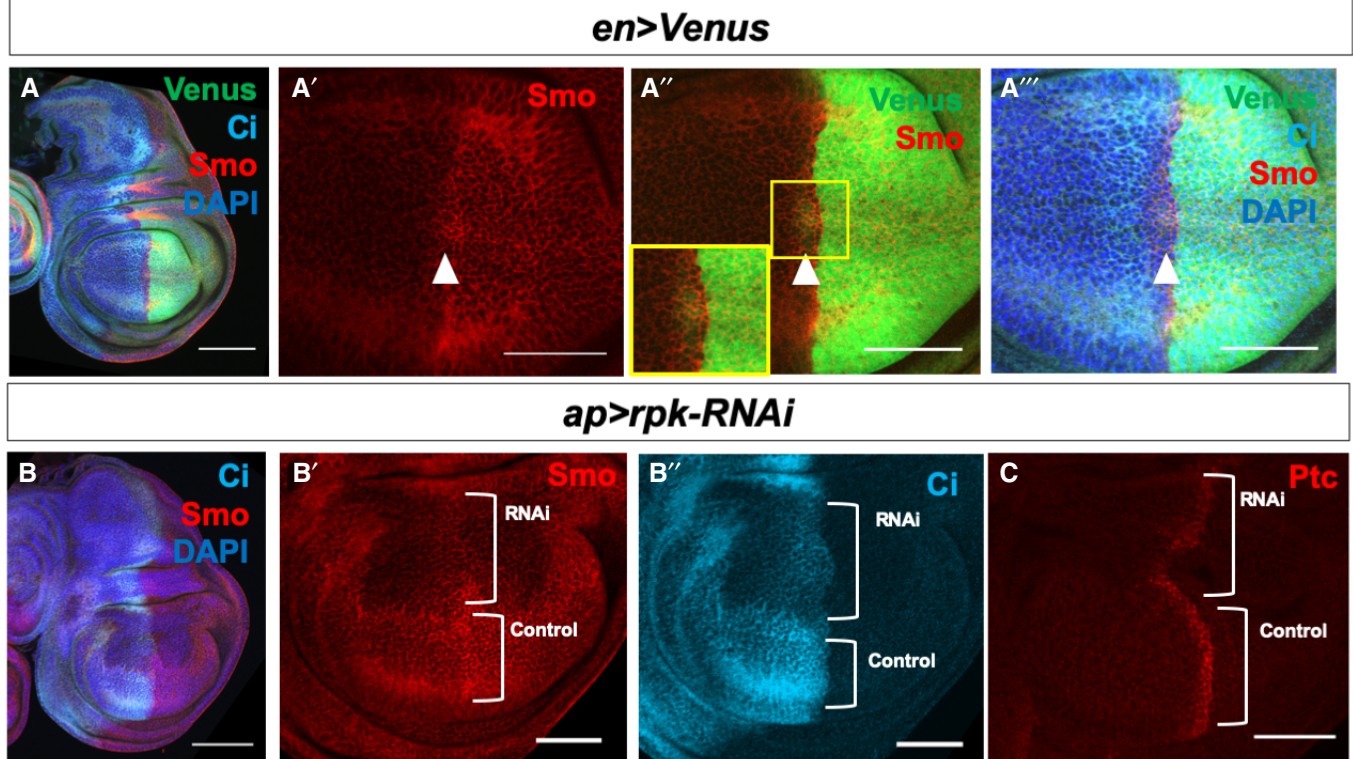

**Figure 4.  The ENaC channel Rpk is required for Hh signal transduction.**

A–A‴   Immunostaining of Smo protein (red) and full-length Ci (light blue) in discs expressing *en-Venus*. Smo is expressed at higher levels in the P compartment and in a stripe 5–10 cells wide just anterior to the A-P compartment boundary. White arrowhead indicates cells just anterior to the A-P compartment boundary.

B, C   Effect of expressing an RNAi against the ENaC channel *rpk* in the dorsal compartment of the disc using *ap-Gal4*. Knockdown of Rpk in the dorsal compartment results in decreased accumulation of the Hh signal transducer Smo (B, B′) and the activator form of Ci (B, B″). The expression of *ptc*, visualized using an anti-Ptc antibody (C), a downstream target gene of Hh signalling, is also diminished. The rpk-RNAi line BL:39053 is used in all panels shown, but the phenotype was validated with a second RNAi line (BL:25847).

Data information: All scale bars are 50 μm, except for (A) and (B), where scale bars are 100 μm.

Wietek *et al*, 2017), resulted in a reduction of Smo on the plasma membrane as well as in internal compartments (Fig 5J–L). This suggests that short-term hyperpolarization generally reduces Smo levels, perhaps due to the fact that inactivated Smo is targeted for proteasome-mediated degradation following internalization, as has been reported previously (Li *et al*, 2012).

### $V_{mem}$ regulates Smo membrane localization independently of Ptc in the wing disc

Since a chemiosmotic mechanism has been hypothesized to drive Ptc-mediated inhibition of Smo (Myers *et al*, 2017), we wondered if depolarization might reduce the inhibitory effect of Ptc on Smo. If this were the case, ectopically depolarizing cells of the posterior compartment of the wing disc should have no effect on Smo membrane localization, since Ptc is not expressed in these cells (Hooper & Scott, 1989; Nakano *et al*, 1989). To test this, we expressed ChR2 in the dorsal compartment of the wing disc and exposed dissected wing discs to 25 min of activating light. Smo membrane abundance was increased in the anterior dorsal compartment of these discs as compared to discs not exposed to light

(Fig 6A–C), indicating that depolarization has a similar effect on Smo membrane localization in the wing imaginal disc as in the salivary gland. A change in the dorsal portion of the posterior compartment was not readily apparent.

To test the effects of sustained depolarization during development, we expressed the red light-activated optogenetic channel ReAChR, which is activatable through the larval cuticle, in the dorsal compartment of the wing disc. Animals were raised on a 12-h light/dark cycle, and imaginal discs were harvested in the third larval instar and stained with an anti-Smo antibody. Under these conditions, membrane abundance of Smo was markedly increased in the dorsal portion of both anterior and posterior compartments in these discs, as compared to the ventral compartments (Fig 6D and D′), indicating that depolarization can mediate increased Smo membrane abundance independently of Ptc. The tissue was also overgrown, in line with other reported phenotypes of Hh pathway hyperactivation (Christiansen *et al*, 2012). In the wing pouch, expression of the Hh target gene *ptc* was increased in the dorsal compartment of discs expressing *ap>ReAChR* (Fig 6E and E′). Thus, optogenetic manipulation of $V_{mem}$ *in vivo* can regulate Smo localization and abundance even in cells that do not express Ptc.

## Discussion

In this study, we show that $V_{mem}$ is patterned in a spatiotemporal manner during development of the wing disc of *Drosophila* and that it regulates Hedgehog signalling at the compartment boundary. First, we have shown that cells immediately anterior to the

compartment boundary are relatively more depolarized than cells elsewhere in the wing pouch. This region coincides with the A cells where Hh signalling is most active, as evidenced by upregulation of Ptc. Second, we found that the expression of at least two regulators of $V_{mem}$, the ENaC channel Rpk and the alpha subunit of the Na$^+$/K$^+$ ATPase are expressed at higher levels in this same

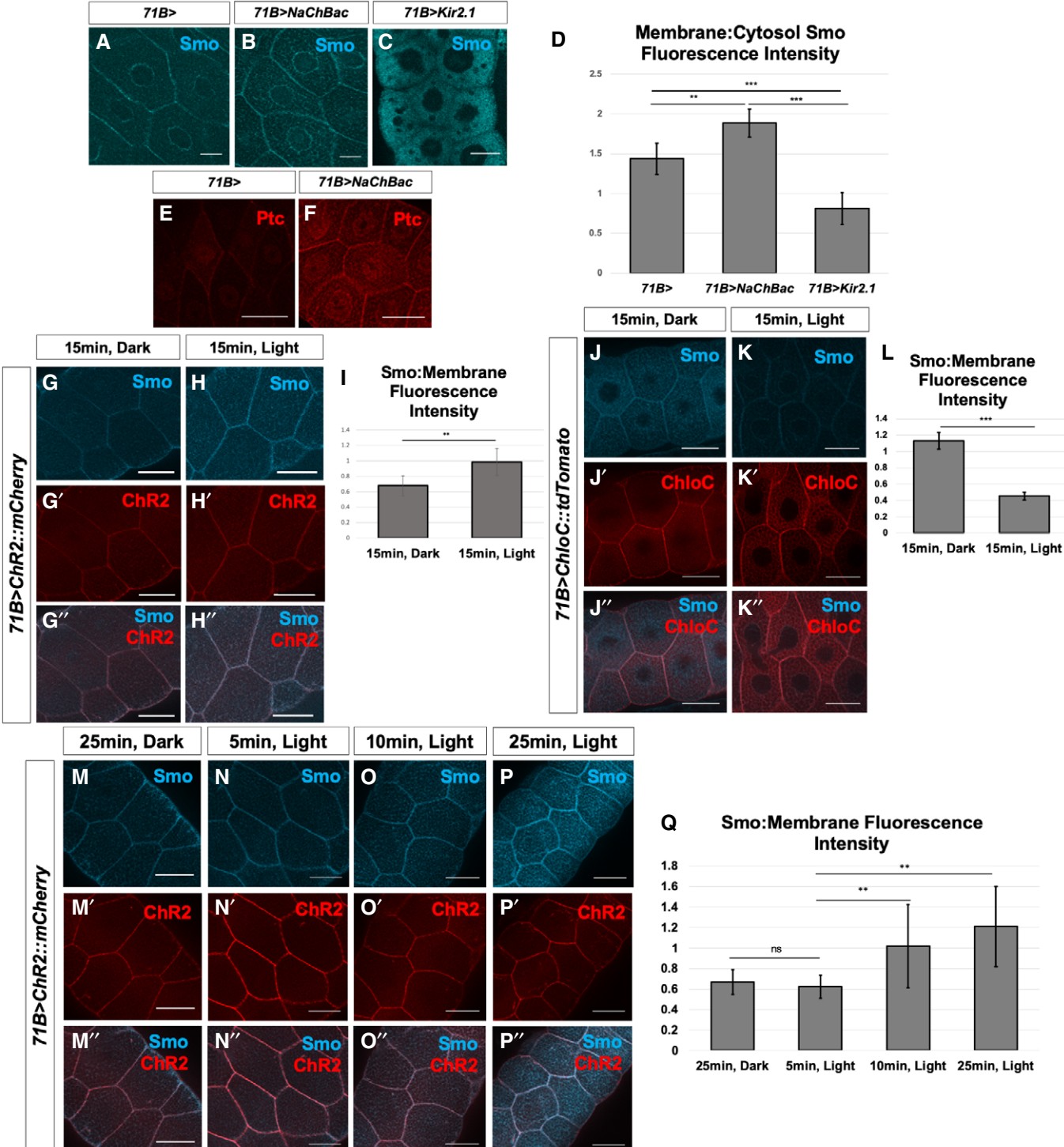

**Figure 5.**

◄

**Figure 5.  Membrane potential modulates Smo membrane abundance and downstream Hh signal transduction in larval salivary glands.**

A–Q   Manipulation of $V_{mem}$ in salivary glands dissected from third-instar larvae. (A–C) Localization of Smo protein (blue) following expression of *71B-Gal4* (A), *71B-Gal4*
      and *UAS-NaChBac* (B) or *71B-Gal4* and *UAS-Kir2.1* (C). Expression of the sodium channel NaChBac increases membrane localization of Smo while expression of the
      potassium channel Kir2.1 reduces membrane localization and increases cytosolic Smo (B, C). (D) Quantification of the ratio of membrane Smo fluorescence to
      cytosol fluorescence. $N = 7$ salivary glands per genotype. Data were compared using ANOVA followed by Tukey's test for significance (** indicates $P < 0.01$, ***
      indicates $P < 0.001$), error bars are standard deviations. (E, F) Expression of the Hh target gene *ptc* is visualized using anti-Ptc immunostaining. Compared to *71B-
      Gal4* (E), Ptc expression is increased in glands expressing *71B-Gal4* and *UAS-NaChBac* (F). (G-I) Effect of expression and activation of the depolarizing
      channelrhodopsin ChR2 on Smo protein membrane abundance. After dissection, glands were kept in either dark conditions (G) or exposed to 15 min of blue light
      (H). (I) Quantification of membrane fluorescence; $N = 13$ glands, data compared using an unpaired *t*-test (** indicates $P < 0.01$, error bars are standard deviations.
      (J–L) Effect of expression and activation of the Cl⁻-selective channelrhodopsin ChloC. An overall reduction in Smo immunostaining was observed following channel
      activation (K), as compared to control glands kept in the dark (J). (L) Quantification of membrane fluorescence; $N = 10$ glands, data were compared using an
      unpaired *t*-test (*** indicates $P < 0.001$), error bars are standard deviations. (M–Q) Time course of change in Smo localization. Salivary glands expressing the
      channelrhodopsin ChR2 were subjected to increasing intervals of activating light, then fixed and stained for Smo protein (M–P"). (Q) Quantification of membrane-
      associated fluorescence; $N = 7$ glands/timepoint, data were compared using ANOVA followed by Tukey's test for significance (** indicates $P < 0.01$), error bars are
      standard deviations. Scale bars are 50 μm, except for (A–C), where scale bars are 25 μm.

portion of the disc. Third, by altering Hh signalling, we demonstrate that the expression of both Rpk and ATPα is increased in cells with increased Hh signalling. Fourth, by manipulating Hh signalling in the disc and using optogenetic methods, both in the salivary gland and wing disc, we show that membrane depolarization promotes Hh signalling as assessed by increased membrane localization of Smo, and expression of the target gene *ptc*. Thus, Hh-induced signalling and membrane depolarization appear to mutually reinforce each other and thus contribute to the mechanisms that maintain the segregation of A and P cells at the compartment boundary.

## Developmentally regulated patterning of $V_{mem}$ in the wing disc

We have observed two regions of increased DiBAC fluorescence in the wing imaginal disc. We did not observe obvious upregulation of Rpk and ATPα in other discs, and therefore, our studies have focused on the region immediately anterior to the A-P compartment boundary in the wing disc. In the late L3 wing disc, we observed a region of increased DiBAC fluorescence in the A compartment in the vicinity of the D-V boundary. This corresponds to a "zone of non-proliferating cells" (ZNC) (O'Brochta & Bryant, 1985). Interestingly, the ZNC is different in the two compartments. In the A compartment, two rows of cells are arrested in G2 while in the P compartment, a single row of cells is arrested in G1 (Johnston & Edgar, 1998). Our observation of increased DiBAC fluorescence in the DV boundary of only the A compartment is consistent with previous reports that cells become increasingly depolarized as they traverse S-phase and enter G2 (Cone, 1969), reviewed in (Blackiston *et al*, 2009). In contrast, cells in G1 are thought to be more hyperpolarized. Additionally, we observed increased expression of the ENaC channel Rpk in two rows of cells at the D-V boundary in the anterior compartment (Fig 2A″), indicating that increased expression of Rpk could contribute to the depolarization observed in those cells. We note, however, that the increased DiBAC fluorescence in these cells was not entirely eliminated by exposing discs to amiloride, indicating that other factors are also likely to contribute.

## How does membrane depolarization relate to Hh signalling?

Our data are consistent with a model where membrane depolarization and Hh-induced signalling mutually reinforce each other in the

cells immediately anterior to the compartment boundary. Both membrane depolarization and the presence of Hh seem necessary for normal levels of activation of the Hh signalling pathway in this region; neither alone is sufficient (Fig 7). First, we have shown that Hh signalling promotes membrane depolarization. We have also shown that the expression of Rpk just anterior to the A-P compartment boundary is dependent upon Hh signalling. Elevated Rpk expression is not observed when a $hh^{ts}$ allele is shifted to the restrictive temperature, and cells become more depolarized when Hh signalling is constitutively activated through expression of the *ci3m* allele. Previously published microarray data suggest that Rpk as well as another ENaC family channel Ppk29 are both enriched in cells that also express *ptc* (Willsey *et al*, 2016). However, there is no antibody to assess Ppk29 expression currently. The sensitivity of the depolarization to amiloride indicates that these and other ENaC channels make an important contribution to the membrane depolarization.

Second, we have shown that the depolarization increases Hh signalling. The early stages of Hh signalling are still incompletely understood (reviewed by Petrov *et al*, 2017). Hh is thought to bind to a complex of proteins that includes Ptc together with either Ihog or Boi. This alleviates an inhibitory effect on Smo, possibly by enabling its access to specific membrane sterols. Interestingly, it has recently been proposed that Ptc might function in its inhibitory capacity by a chemiosmotic mechanism where it functions as a Na⁺ channel (Myers *et al*, 2017). An early outcome of Smo activation is its localization to the membrane where its C-terminal tail becomes phosphorylated and its ubiquitylation and internalization are prevented (Zhang *et al*, 2004; Li *et al*, 2012). By manipulating channel expression in the wing disc, and by optogenetic experiments in both the salivary gland and wing disc, we have shown that membrane depolarization can promote Hh signalling as assessed by increased Smo membrane localization and increased expression of the target gene *ptc*. The time course of Smo activation is relatively rapid (over minutes) and is therefore unlikely to require new transcription and translation. In the P compartment, membrane Smo levels are elevated likely because of the complete absence of Ptc, and some downstream components of the Hh signalling pathway are known to be activated (Ramírez-Weber *et al*, 2000). However, since Ci is not expressed in P cells, target gene expression is not induced. In the cells just anterior to the boundary, the partial inhibition of Ptc by Hh together with membrane depolarization seem to combine to achieve similar

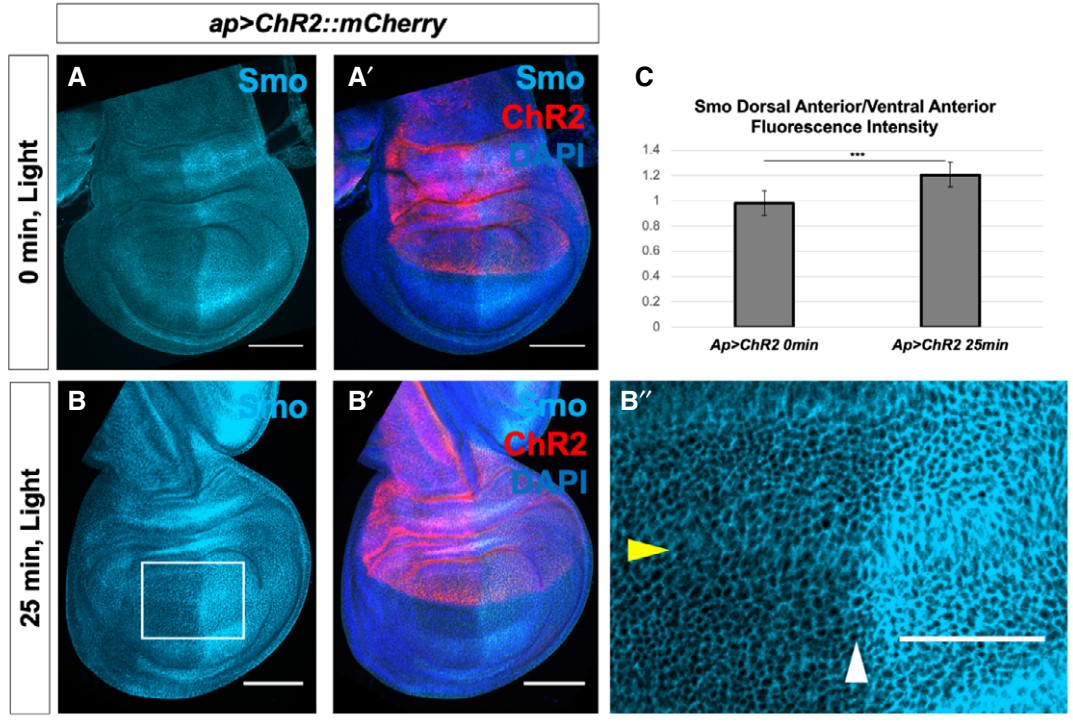

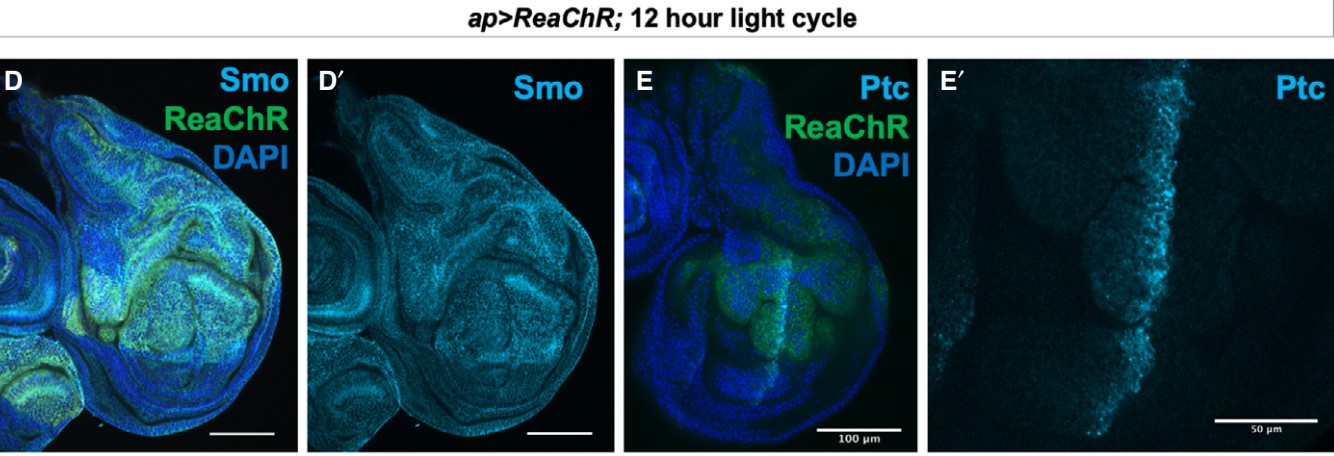

**Figure 6.** $V_{mem}$ regulates Smo membrane localization independently of Ptc in the wing imaginal disc.

A–B″  Wing discs expressing the channelrhodopsin ChR2 were subjected to 0 min or 25 min of activating light in culture, then fixed and stained for Smo protein. In (B″), the yellow arrowhead indicates dorsoventral compartment boundary, and the white arrowhead indicates anteroposterior compartment boundary.

C  Quantification of the ratio of mean Smo fluorescence intensity in 25 μm² regions in the dorsal anterior compartment compared to the ventral posterior compartment. $N = 5$ for 0 min discs, $n = 9$ for 25 min discs, data were compared using an unpaired t-test (***$P < 0.001$), error bars are standard deviations. Identical analyses were carried out comparing dorsal posterior fluorescence to ventral posterior fluorescence ($P = 0.132$), dorsal posterior fluorescence to dorsal anterior fluorescence ($P = 0.12$) and ventral posterior fluorescence to ventral anterior fluorescence ($P = 0.36$).

D–E′  Discs expressing the red light-activated channelrhodopsin ReaChR under control of ap-Gal4 were raised on a 12-h light/dark cycle, dissected and stained for Smo protein (D, D′) or Ptc protein (E, E′).

Data information: Scale bars are 100 μm in all panels except (B″) and (E′), where scale bars are 50 μm.

levels of Smo membrane localization. More anteriorly, the absence of this mutually reinforcing mechanism appears to result in Smo internalization (Fig 7).

Our experiments do not point to a single mechanism by which depolarization promotes Hh signalling. It is possible that

depolarization results in increased $Ca^{2+}$ levels by opening $Ca^{2+}$ channels at the plasma membrane or by promoting release from intracellular sources (e.g. the ER or mitochondria). Indeed, there is evidence that $Ca^{2+}$ entry into the primary cilium promotes Hh signalling, and recent work shows that targets of Sonic Hedgehog

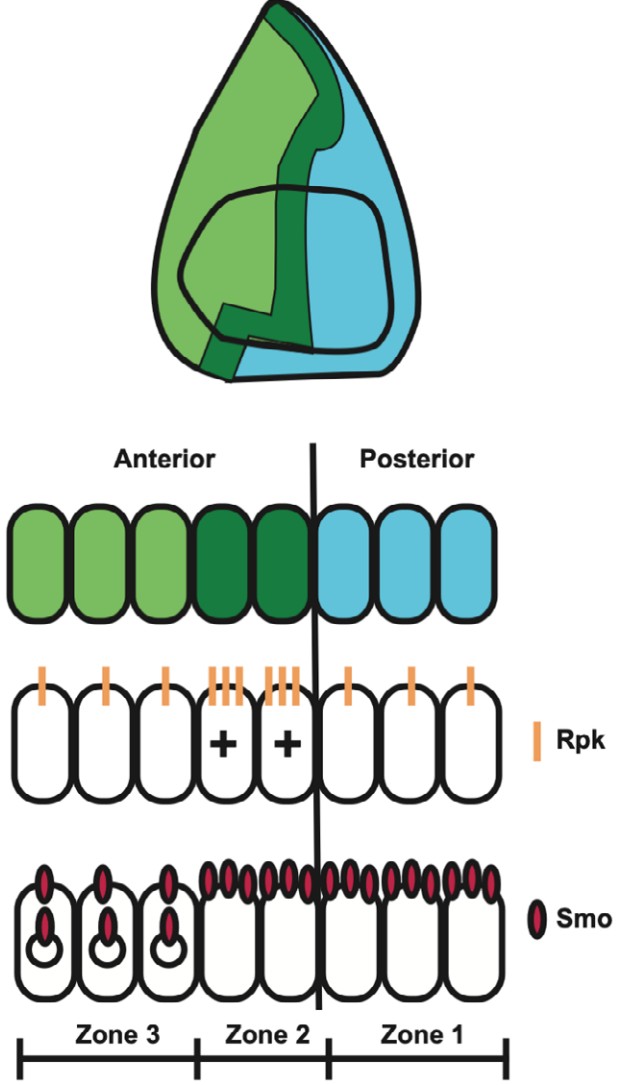

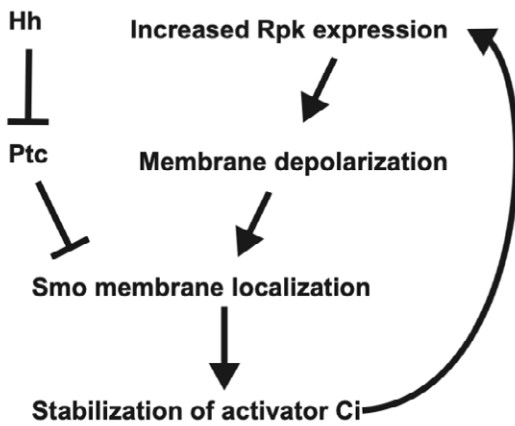

**Figure 7. Membrane potential modulates Hh signalling at the anteroposterior compartment boundary.**

Cells in the posterior compartment (blue) produce the short-range morphogen Hedgehog, which is received by cells immediately anterior to the compartment boundary, in the anterior compartment (dark green). Hh signal releases Ptc-mediated inhibition of Smo, allowing for the stabilization of the transcriptional co-effector Ci, and increased expression of the ENaC channel Rpk, which depolarizes cells immediately anterior to the compartment boundary. Depolarization results in increased membrane abundance of the Hh activator Smo and increased transduction of Hh signal. This, in turn, increases expression of Rpk, reinforcing high levels of Hh signal transduction in the region immediately anterior to the compartment boundary.

(Shh) signalling during mammalian development is augmented by $Ca^{2+}$ influx (Delling *et al*, 2013; Klatt Shaw *et al*, 2018). A second possibility is that membrane depolarization could, by a variety of mechanisms, activate the kinases that phosphorylate the C-terminal tail of Smo and maintain it at the plasma membrane in an activated state. Depolarization could also impact electrostatic interactions at the membrane that make the localization of Smo at the membrane more favourable. Since Rpk and ATPα are expressed at higher levels in the cells that receive Hh, which have been postulated to make synapse-like projections with cells that produce Hh (González-Méndez *et al*, 2017), it is conceivable that these channels could modulate synapse function. Additionally, while our work was under review, it has been reported that reducing glycolysis depletes ATP

levels and results in depolarization in the wing imaginal disc, reducing the uptake of Hh pathway inhibitors and stabilizing Smo at the cell membrane (Spannl *et al*, 2020). Importantly, all these mechanisms are not mutually exclusive and their roles in Hh signalling are avenues for future research.

It is now generally accepted that both cell–cell signalling and mechanical forces have important roles in cell fate specification and morphogenesis. Our work here adds to a growing body of literature suggesting that changes in $V_{mem}$, a relatively understudied parameter, may also have important roles in development. Integrating such biophysical inputs with information about gene expression and gene regulation will lead to a more holistic understanding of development and morphogenesis.

# Materials and Methods

### *Drosophila* strains and husbandry

Animals were raised on standard medium as used by the Bloomington *Drosophila* Stock Center. All animals were raised at 25°C, except *ap>rpk-RNAi* crosses, which were raised at 18°C to reduce lethality. Crosses using GAL80$^{TS}$ and the *hh$^{TS2}$* allele were raised at 18°C and then upshifted to 30°C at the reported timepoints.

Stocks used in this study include the following: *hsFLP;; act<stop<Gal4, UAS-RFP/S-T, yw;ap-Gal4/Cy0; TM2/TM6B,;;ap-Gal4,;ptc>RFP;, w;;UAS-Ci3 M,;; 71B-Gal4*, and *w;UAS-Kir2.1; UAS-Kir2.1/TM6B* (from Kristin Scott, UC Berkeley, USA).

Stocks obtained from the Bloomington Stock Center (Bloomington, IN, USA) include *UAS-ArcLight* (BL:51056), *UAS-rpk-RNAi;* (BL:39053, 25847—data using BL:39053 are shown and confirmed with BL:25847), *UAS-NaChBac;* (BL:9466), *UAS-ChR2::mCherry;* (BL:28995), *UAS-ReaChR::Citrine;* (BL:53741), *UAS-ChloC::tdTom* (BL:76328), *en-Gal4;* (BL:6356),; *UAS-Ptc-RNAi;* (BL:55686).

### Live imaging and optogenetics

Larvae were washed with 70% EtOH and PBS prior to dissection. Live tissue was dissected in Schneider's media (#21720001, Gibco), and care was taken to not damage or stretch tissue. For DiBAC staining, imaginal discs were incubated in 1.9 μM DiBAC$_4$(3) (bis-(1,3-dibutylbarbituric acid) trimethine oxonol; DiBAC$_4$(3); Molecular Probes)) in Schneider's media for 10 min with gentle rotation. A small amount media was used to mount the discs, such that the addition of a coverslip did not destroy the tissue, and discs were imaged right away. 100 μM amiloride (#A7419, Sigma-Aldrich) and 100 μM ouabain (#O3125, Sigma-Aldrich) were added to DiBAC solution for pharmacology experiments. Discs imaged in FM4-64 dye (#T13320, Thermo Fisher) were incubated in 9 μM FM4-64, and imaged without washing, to preserve staining of the cell membrane.

For optogenetics experiments in the salivary gland, carcasses were dissected and cleaned (fat body removed) in Schneider's medium. Carcasses were loaded onto a glass cover slip in a large drop of Schneider's, and either kept in the dark (control condition) or exposed to activating light (480 nm for ChR2 and ChloC experiments, 647 nm for ReaChR experiments). After exposure, carcasses were immediately fixed and prepared for immunohistochemistry. For optogenetics experiments using the ChR2 channel in the wing imaginal disc, carcasses were dissected and cleaned (fat body removed) in Schneider's medium, then either fixed (0 min activating light condition), or loaded onto a glass cover slip in a large drop of Schneider's, and exposed to 25 min of activating light (480 nm). Following activation, discs were fixed and prepared for immunohistochemistry. For experiments using the ReaChR channel in the wing imaginal disc, flies were raised on a 12-h light/dark cycle, and imaginal discs were harvested and prepared for immunohistochemistry during the third larval instar.

### Immunohistochemistry

Imaginal discs were dissected in phosphate-buffered saline, fixed for 20 min in 4% PFA at room temperature, permeabilized in phosphate-buffered saline with 0.1% Triton X-100, and blocked in 10% Normal Goat Serum. Primary antibodies used were: rat anti-Ci (1:10, #2A1; Developmental Studies Hybridoma Bank, DSHB), mouse anti-Smo (1:10, #20C6; DSHB), rabbit anti-Rpk (1:500) (gift from Dan Kiehart), mouse anti-ATPα (1:100, #a5; DSHB), mouse anti-Cut (1:100, #2B10; DSHB), mouse anti-Wg (1:100, #4D4 DSHB), and mouse anti-Ptc (1:50, #Apa1; DSHB). Secondary antibodies used were: goat anti-rabbit 488 (#A32731; Invitrogen), goat anti-mouse 488 (#A32723; Invitrogen), goat anti-mouse 555 (#A32727; Invitrogen), goat anti-rat 555 (#A-21434; Invitrogen), goat anti-mouse 647 (#A32728; Invitrogen), goat anti-rabbit 647 (#A32733; Invitrogen) and goat anti-rat 647 (#A-21247; Invitrogen). Nuclei were stained with DAPI (1:1,000, Cell Signaling). Samples were imaged on a Zeiss Axio Imager.M2 with Apotome.2.

### Quantification and statistical analysis

Fluorescence intensity measurements were recorded using FIJI software (NIH, Bethesda, USA). *P* values were obtained using ANOVA and unpaired Student's *t*-tests (GraphPad). Error bars in all graphs are standard deviation. *P* value significance < 0.001: ***; 0.001 to 0.01: **; 0.01 to 0.05: *; > 0.05: not significant.

### Mosaic tissue generation

To generate clones expressing *ci3m* or *ptc$^{RNAi}$*, *hsFLP;;act<STO-P<UAS-RFP/S-T* virgin females were crossed to *UAS-Ci3m* or *UAS-Ptc$^{RNAi}$* males. Larvae were collected as described above, and vials were subjected to a 10 min heat shock in a 37°C water bath 48 h before dissection and live imaging in DiBAC, or fixation and preparation for immunohistochemistry.

# Data availability

This study includes no data deposited in external repositories.

Expanded View for this article is available online.

## Acknowledgements

We are indebted to Riku Yasutomi, who conducted related experiments that helped shape our thinking about this work, other members of the Hariharan laboratory, Ehud Isacoff, Diana Bautista, David Bilder, Kristin Scott, Henk Roelink, Michael Levin, Evan Miller, and Julia Lazzari-Dean for helpful discussions and feedback. We also thank Diana Bautista, David Bilder, Kristin Scott, and Dan Kiehart for reagents and fly stocks. This work was funded by NIH grant R35 GM122490.

## Author contributions

ME-B and IKH conceived the study. ME-B designed experiments, collected and analysed data, and prepared figures. ME-B and IKH wrote the manuscript.

## Conflict of interest

The authors declare that they have no conflicts of interest.

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
