## [Review Process File · EMBO Reports]

Membrane potential regulates Hedgehog signaling in the *Drosophila* wing imaginal disc

Maya Emmons-Bell and Iswar Hariharan
DOI: [10.15252/embr.202051861](https://doi.org/10.15252/embr.202051861)

Corresponding author(s): Iswar Hariharan (ikh@berkeley.edu)

Review Timeline:

Submission Date:	9th Oct 20
Editorial Decision:	19th Oct 20
Revision Received:	15th Nov 20
Editorial Decision:	22nd Dec 20
Revision Received:	29th Dec 20
Accepted:	15th Jan 21

Transaction Report: Please note that the manuscript was previously reviewed at another journal and the reports were taken into account in the decision making process at EMBO Reports. Since the original reviews are not subject to EMBO Press' transparent review process policy, the reports and author response cannot be published.

Dear Iswar,

Thank you for the submission of your research manuscript to EMBO reports. It had been reviewed for another journal and you have submitted the referee reports alongside your manuscript.

I have read your manuscript and the referee reports. I think it is clear that the referees consider your data on the patterned depolarization of cells across the A/P boundary and the positive feedback loop on Hh signalling intriguing. It is however also clear that the data appear preliminary and that a significant amount of work will be required to strengthen the data. You indicated that you have already performed additional experiments to substantiate the conclusions on the Hh-controlled expression of Rpk and the depolarization of cells anterior to the compartment boundary and how depolarization in turn promotes Hh signaling. You also indicated that you plan to remove the data on competitive cell survival and I agree with these points. It will certainly be most important to strengthen the links between Hh signaling and membrane depolarization. I feel however that data on whether the depolarization of anterior cells has an impact on A/P boundary formation would certainly further strengthen the study, at least it should be discussed in depth.

As discussed, we would like to invite you to revise your manuscript for potential publication in EMBO reports. Please address all referee concerns in a complete point-by-point response. Your revised manuscript will be assessed by the same set of referees who evaluated the first version and acceptance will depend on a positive outcome of this second round of review. It is EMBO reports policy to allow a single round of revision only and acceptance or rejection of the manuscript will therefore depend on the completeness of your responses included in the next, final version of the manuscript.

*** Temporary update to EMBO Press scooping protection policy:

We are aware that many laboratories cannot function at full efficiency during the current COVID-19/SARS-CoV-2 pandemic and have therefore extended our 'scooping protection policy' to cover the period required for a full revision to address the experimental issues highlighted in the editorial decision letter. Please contact the scientific editor handling your manuscript to discuss a revision plan should you need additional time, and also if you see a paper with related content published elsewhere.***

- 1) A data availability section is missing.
- 2) Your manuscript contains error bars based on $n=2$. Please use scatter blots showing the individual datapoints in these cases. The use of statistical tests needs to be justified.

Please note that for all articles published beginning 1 July 2020, the EMBO Reports reference style will change to the Harvard style for all article types. Details and examples are provided at <https://www.embopress.org/page/journal/14693178/authorguide#referencesformat>

2) individual production quality figure files as .eps, .tif, .jpg (one file per figure).

Please download our Figure Preparation Guidelines (figure preparation pdf) from our Author Guidelines pages

<https://www.embopress.org/page/journal/14693178/authorguide> for more info on how to prepare your figures.

4) a complete author checklist, which you can download from our author guidelines (). Please insert information in the checklist that is also reflected in the manuscript. The completed author checklist will also be part of the RPF.

5) Please note that all corresponding authors are required to supply an ORCID ID for their name upon submission of a revised manuscript (). Please find instructions on how to link your ORCID ID to your account in our manuscript tracking system in our Author guidelines
()

6) We replaced Supplementary Information with Expanded View (EV) Figures and Tables that are collapsible/expandable online. A maximum of 5 EV Figures can be typeset. EV Figures should be cited as 'Figure EV1, Figure EV2" etc... in the text and their respective legends should be included in the main text after the legends of regular figures.

7) Please note that a Data Availability section at the end of Materials and Methods is now mandatory. In case you have no data that requires deposition in a public database, please state so instead of refereeing to the database.

See also < <https://www.embopress.org/page/journal/14693178/authorguide#dataavailability>>).

Please note that the Data Availability Section is restricted to new primary data that are part of this study.

8) We would also encourage you to include the source data for figure panels that show essential data. Numerical data should be provided as individual .xls or .csv files (including a tab describing the data). For blots or microscopy, uncropped images should be submitted (using a zip archive if

multiple images need to be supplied for one panel). Additional information on source data and instruction on how to label the files are available .

10) Regarding data quantification

- the name of the statistical test used to generate error bars and P values,
- the number (n) of independent experiments (please specify technical or biological replicates) underlying each data point,
- the nature of the bars and error bars (s.d., s.e.m.)

11) As part of the EMBO publication's Transparent Editorial Process, EMBO reports publishes online a Review Process File to accompany accepted manuscripts. This File will be published in conjunction with your paper and will include the referee reports, your point-by-point response and all pertinent correspondence relating to the manuscript.

I look forward to seeing a revised version of your manuscript when it is ready. Please let me know if you have questions or comments regarding the revision.

Kind regards,

Martina

Martina Rembold, PhD
Editor
EMBO reports

We thank the reviewers for their careful reading of the manuscript and for the many constructive suggestions for improvement. Rather than re-submit to *eLife*, we have chosen to transfer the manuscript to *EMBO Reports*. This decision was prompted by the recent publication in *The EMBO Journal* of a paper from the laboratory of the late Dr. Suzanne Eaton – Spannll et al., 2020 (<https://www.embopress.org/doi/full/10.15252/emboj.2019101767>) In that manuscript, the authors primarily investigate the effect of altering the expression of glycolytic enzymes in the wing disc and thus altering ATP levels. They show that a reduction in ATP levels appears to promote Hedgehog signaling. They show, using the same voltage-reporting dye that we use (DiBAC) that reducing ATP levels increases membrane depolarization and increased membrane localization of Smoothed and they attribute some of this effect to a reduced uptake of N-acylethanolamides which can function as Hedgehog-pathway inhibitors. Their work did not address the endogenous pattern of membrane depolarization, patterned expression of ion channels or the effect of manipulating membrane potential on Hedgehog signaling.

We believe that our work is complementary to the findings of Spannll et al and think therefore that it is important that our work is published relatively soon. Please note that the version of our manuscript that was submitted to *eLife* was concurrently deposited in *BioRxiv* (June 2, 2020). In their response to reviews (downloadable from the EMBO J website) dated June 10, 2020, Spannll et al. discuss the work from our preprint in response to a reviewer's question about whether manipulating membrane potential can affect Smoothed localization (they did not do this experiment themselves but point to our work). Our, preprint, however was not discussed or cited in the final published version of their manuscript. Thus it is clear that our findings were made independently of theirs. The Editor of *EMBO Reports*, Dr. Martina Rembold is willing to consider our manuscript for *EMBO Reports* subject to re-review.

When our manuscript was reviewed for *eLife*, the reviewers found the findings intriguing but also commented on the preliminary nature of some of the observations. Together the four reviewers suggested many additional experiments ranging from the inclusion of additional controls to whole new lines of experiments that involved the generation of loss-of-function mutants. We want to emphasize that we are still conducting research under low-density conditions because of the Covid pandemic and believe that some of the suggested experiments could take us a year or more. However, in response to the critiques, we have removed the more preliminary portions of the manuscript, incorporated many of the suggestions of the reviewers (especially the inclusion of appropriate controls) and focused the manuscript on the link between membrane potential and Hedgehog signaling. Importantly, we have added new experimental data where we show in the wing disc that optogenetic manipulation of membrane potential can modulate Hedgehog signaling.

Importantly we have solidified our four main conclusions each of which does not appear in the Spannll et al. paper:

- 1) We show that membrane potential is spatially patterned in the wing disc and that the region immediately anterior to the compartment boundary is more depolarized than other regions in the wing disc.
- 2) We show that certain ion channels, specifically *ripped pocket (rpk)* and *ATP-a* are expressed at elevated levels in this region
- 3) Hh pathway signaling is necessary for altered expression of these ion channels and interrupting Hh signaling reduces their expression in this region and membrane depolarization
- 4) We directly manipulate membrane potential in salivary gland cells and the wing disc using optogenetic methods and show that depolarizing the membrane increases membrane localization independently of Patched. (The optogenetic experiments in the wing disc, done both in culture and in vivo, were not in the previous submission).

Taken together our data show that Hh signaling and membrane depolarization are mutually reinforcing.

We have removed the sections that deal with the role of depolarization in maintaining the compartment boundary and on the compartment-specific survival of clones with altered membrane potential. We agree with the reviewers that those portions of the manuscript would benefit from more detailed investigations.

We therefore present a point-by-point response to the comments made by each reviewer to the original version of the manuscript. We hope that the reviewers can assess the merits of the requested changes in the context of the need to publish our work not long after the paper of Spannl et al.

Reviewer 1:

*In this work Emmons-Bell and colleagues describe patterning of membrane potential (*V_{mem}*) within epithelial cells and its effects on the Hedgehog (Hh) signaling pathway in the developing *Drosophila* wing. This work reveals an important aspect of cell-cell communication in epithelial tissues during development. *V_{mem}* is traditionally studied in the nervous system and it is key during synaptic transmission. Here the authors demonstrate that membrane depolarization is required for Hh signaling activity and how this signaling feeds back into the membrane potential by regulating ion channels.*

*The work is a step ahead in the research of the mechanisms regulating cell signaling during development of non-neuronal tissues. It is disappointing that the paper does not go in detail at which level *V_{mem}* might affect Hh signaling but it simply presents the findings in a descriptive manner. However, many of the findings are exciting and open new avenues for further studies. In*

general it is a good quality, timely topic research and the manuscript is well written. There are some issues that the authors should address before I can recommend its publication.

We thank the reviewer for their comments and constructive criticism.

Main comments:

1) A secondary aspect derived of this work is the effect of Vmem on the maintenance of the anterior/posterior compartment border. In my opinion, the A/P border has to necessarily be affected by depolarization of the A compartment since it is known that Hh signaling maintains the lineage restriction at the AP compartment border. However, the mechanism of a direct implication of Vmem in the maintenance of the lineage restriction has not been analyzed or even discussed. Therefore, I consider that this result should not be part of the title.

It has also been demonstrated that Ihog (Hsia et al., 2017) contributes to the formation of the A/P compartment border independently of its role as coreceptor of the Hh pathway. This previous results should be also discussed.

We have removed the data concerning A/P boundary maintenance, and altered the title of the paper accordingly.

We have added discussion of Ihog and its contributions to boundary maintenance (Lines 51-63).

2) To better analyze the function of Rpk in Vmem and Hh signaling, a temporal control of RNAi-mediated Knockdown of Rpk by Gal80ts should be done in some experiments. Also available from Bloomington there is a gRNA expressing stock that permits tissue specific rpk knockout (Port et al., 2020).

We have included experiments in which we did a temporally-controlled knockdown of *rpk* with *Gal80ts* in the wing pouch to show that Rpk contributes to depolarization in the wing imaginal disc (Fig EV1).

3) The authors do not prove any functional link between Rpk and Vmem. Also, it is not solved at which level Vmem might affect Hh signaling. This missing link seems crucial and has to be solved. It is very well possible, and the authors mention it in the discussion, that membrane potential affects Hh signaling at different levels. The structure of Ptc has been shown to have high similarity with the RND superfamily of transporters. These transporters use the energy of an ion gradient to move small molecules across the membrane. Ptc specifically requires extracellular Na⁺ to regulate Smoothed. The authors could test whether membrane polarization could affect this receptor membrane positioning. This might be the key to understand Smo regulation via Ptc.

We present two lines of evidence that the effects we observe on Smo localization are independent of Ptc. First, expression of *rpkRNAi* using *ap-Gal4* reduces Smo level in the dorsal part of the wing pouch in both anterior and posterior compartments. Similarly, depolarization using optogenetic methods both in discs in culture and in vivo increases Smo levels and membrane localization in both anterior and posterior compartments. Thus we can conclude that at least in the posterior compartment (where *ptc* is not expressed) Ptc function is not necessary for depolarized V_{mem} to regulate membrane accumulation of Smo.

4) To further confirm Hh signaling dependence additional experiments using alternative tools to block the Hh pathway, such as Dispatched down-regulation or smo- clones can be done. In the same line of thought, analysis of the effects of Hh over-expression or ptc- clones are also desirable adding up to constitutively active Ci clones.

In the revised manuscript we show the effect of generating constitutively active Ci clones as well as clones where *ptc* has been knocked down (Figure 3). Importantly, we show that clones expressing *ci3m* are depolarized in both A and P compartments while clones expressing *ptcRNAi* are depolarized only in the A compartment consistent with the expression domain of *ptc*.

5) To date there are some recent publications describing Hh reception as a synaptic-like process and thus, the authors should at least discuss the potential effects over this proposed Hh-reception mechanism.

We have included discussion of synaptic-like processes in Hh reception to the text (Lines 413-415).

Reviewer #1 (Minor Comments):

Minor points

- Which is the expression pattern of Rkp in other discs? Does rkp loss of function affect Hh signaling in other discs? This should be addressed or at least discussed.*

Patterned expression of Rpk is not obvious in other discs. We have added discussion of this issue (Lines 348-349).

- The focus of the research is on Hh signaling and some explanation is given about the potential causes of membrane depolarization across the dorsal/ventral compartment. However, it would be useful if the authors could show if there are any signaling consequences over the Wingless pathway or at least discuss it.*

We have added data exploring the effect of *rpk* knockdown on the Wingless pathway. We find that knockdown of Rpk in the wing pouch results in a modest reduction in anti-Wg staining in

this tissue (Fig EV4). More obvious is a dramatic reduction of the target gene Cut (Please note that Cut expression outside the domain of expression of the RNAi and in the myoblasts is not affected). Thus, the effects of manipulating Rpk expression are not restricted to Hh signaling.

Comments on the text:

Line 46. Authors should specify :...imaginal disc, that "at the end of the third larval instar" is composed of... orthat "in mature disc" is composed of.....

Added (Line 43).

Line 49. It would be more suitable to cite original articles than citing the review Blair, 2003.

A citation of the original article describing compartmentalization in *Drosophila* (Garcia-Bellido et al., 1973) has been added (Line 46).

Line 63. Cytonemes cannot be considered "processes" , they are filopodia-like protrusions. References for this sentence should be added.

The text has been changed and references added (Line 62).

Lines 113, 132, 137. It may help if authors do not use the term "dorsoventral stripe" to refer to a stripe of cells that follows the anteroposterior boundary, as many researchers would confuse it with the dorsoventral boundary (going strictly perpendicular to what the authors mean).

We have reworded our description throughout the text to avoid this ambiguity.

Line 134. The patterns of ArcLight and DiBAC are different (Arclight being much more complex) but both show a similar reduction in the AP border. A sentence such as"similar depolarisation pattern in the anteroposterior border".....would be more suitable.

We have found that the fluorescence from ArcLight to be highly variable. We have therefore removed those images from the revised manuscript.

Line 153. The authors should specify the panel when referring to "Figure 1 and Figure Supplementary 1"

Line 161. The title of this section "Altered expression of endogenous ion channels anterior to the compartment boundary" does not seem to describe the content of the section. It would be better "Expression and function of endogenous ion channels in the wing disc"

We have changed the section title to that suggested (Line 136).

Line 262. It is difficult to understand why Smo subcellular localization is reduced after Rpk RNAi x Ap Gal4 expression in the whole dorsal compartment while Rpk is not expressed or barely expressed outside the A/P or D/V compartment borders.

Staining for Rpk is low, but detectable, in all cells in the wing imaginal disc, which could explain why knockdown of the channel affects all cells of the dorsal compartment.

Line 434 It is "Boi" instead of "Doi"

Changed (Line 385).

Reviewer #1 (Additional data files and statistical comments):

Comments on the figures:

Figure 1.

Control for panel F and G would help the reader to realize what and how large is the effect of ouabain treatment in DiBAC general levels.

The appropriate control has been added (Fig. 1F). However, as DiBAC reports only relative changes in Vmem within a tissue, we have tried to minimize making quantitative comparisons between different preparations.

Panel I mentions a dorso/ventral stripe of cells with decreased fluorescence, but this should be an anterior/posterior stripe of cells, visualized on the dorsal compartment.

In addition, a staining with an A/P compartment border marker should be shown.

This data showing ArcLight fluorescence have been removed.

Panel J. To avoid confusion, the authors can just write DIBAC instead of DIBAC4.

Done.

Figure 2.

Control for panel B is recommended.

Ptc stripe seems to be affected (not expected from such a short treatment) suggesting that the disc integrity is affected either by the treatment or by inappropriate manipulation. Do the authors have a clearer example?

We have replaced these panels with a sample that does not have altered Ptc stripe integrity (Fig. 2B-B’’).

Panel D requires control (without amiloride and processed in parallel).

The appropriate control (vehicle and processed along with amiloride-treated and ouabain-treated discs) is available in Fig. 1F, and this information has been added to the figure legend.

Panel E. Is it possible to present an image with better resolution? The authors indicate that the morphology of the disc is altered but they should have disrupted the expression of Rpk in a temporal manner by including tub-Gal80ts to see the influence on Vmem without affecting the size and the shape of the disc.

These images have been removed, and we have included DiBAC staining of a disc expressing *Rpk-RNAi* in the wing pouch using *rn-Gal4* (Fig EV1), and as well as DiBAC staining of a disc expressing *Rpk-RNAi* just anterior to the compartment boundary using *dpp-Gal4* (Fig. 2D-E) to show that *Rpk* contributes to depolarization. In both cases Gal80ts was used to reduce the deleterious effects of long-term knockdown.

Figure 3.

There is not a proper control of Hhts-2 experiment without temperature shift.

The control has been added (Fig. EV2).

Panel F. Neither the up regulation nor the DiBAC endogenous pattern are clearly visible. The cells up regulating DiBAC fluorescence seem to be only in the A compartment. A better replica should be shown. The experiment is anyway not critical, but the claim of the authors is not visible.

We have added a more clear inset illustrating increased DiBAC staining, an image of a whole disc showing many clone examples, and quantification of the variability in clone staining (Fig.3F and Fig. EV3). In Fig EV3 we show that *ci3m*-expressing clones show depolarization in both compartments, and that *ptcRNAi*-expressing clones are only depolarized in the anterior compartment.

Panel G does not give additional information, the image quality is not very good.

This image has been removed.

Figure 4.

Panel B,B'. It is difficult to understand why Smo subcellular localization is reduced in the whole dorsal compartment after Rpk RNAi X Ap Gal4 expression if Rpk is not expressed or barely expressed outside the A/P or D/V compartment border.

Anti-Rpk staining shows low levels of expression in all cells in the wing disc.

Panel C. The tissue is severely affected by this treatment. Could the authors temporally knockdown Rpk using tubGal80ts to allow a proper tissue development and better show the changes?

Although the tissue is affected, we believe the difference in Ptc staining is clear. We have also included anti-Ptc staining of a disc in which Rpk has been knocked down for 48h in the pouch using *rn-Gal4* and *tubGal80ts* to further demonstrate the reduction in Ptc protein following Rpk knockdown (Fig. EV1)

Panel D, D' At least with this quality of images, it is not possible to see a change in Smo localization upon Irk1.RNAi expression. The authors should outline the DV boundary (or coexpress UASGFP) to see whether or not there is a non-autonomous effect via Hh from the wild type ventral side.

We agree. We have removed the experiments that used Irk1.RNAi.

Figure 5.

It would be interesting to analyze Hh signaling (Ptc and Ci expressions), not only the subcellular localization of Smo in salivary glands, but also in the wing disc by the ectopic expression of the bacterial sodium channel NaChBac and the mammalian potassium channel Kir2.1.

We have added experiments in which Smo level and localization as well as Ptc expression are assessed after optogenetic depolarization either in disc culture or in vivo. sustained depolarization with the ReaChR optogenetic channel (Fig. 6). The effects on Smo are obvious. The effect on Ptc is more subtle. These experiments confirm that our observations in the salivary gland also apply to the wing disc.

Figure 6.

The expression of Rpk-RNAi seems to induce apoptosis at the A/P compartment boundary based on the images shown in panels B, B', B'. However, it has been described that the effects on A/P compartment border by abrogating Hh signaling does not provoke cell death. Caspase 3 staining is recommended to analyze cell death and, in case of positive staining, co-expression of Rpk RNAi with p35 is recommended to block apoptosis.

We have focused our revised manuscript on the relationship between membrane potential and Hedgehog signaling. We agree that the effects on compartment boundary maintenance require more investigations.

Reviewer #2 (General assessment and major comments (Required)):

The manuscript by Emmons-Bell reports novel and interesting data indicating a role for membrane potential in Hedgehog signal transduction - and conversely a role for Hedgehog signal transduction in controlling membrane potential, thus providing a potential feedback. This feedback between Hedgehog signaling (or in general cell signaling) and membrane potential is intriguing. However, there are a number of reservations which dampen my enthusiasm: First, quite a number of pieces of data are unconvincing (some require controls), see 'specific comments', raising concerns about the validity of major conclusions. Addressing these concerns will require major additional work. Second, there are some 'lose ends'. The observation that Rpk knockdown affects boundary shape is interesting, but this data set is not well linked to the remaining experiments. Similar for the experiments reporting clone survival. Third, several questions remain open: does membrane potential only affect Hedgehog pathway activity? Are other pathway activities (Wg,N,Dpp) affected when the levels of ion channels are altered (there seems to be some change in membrane potential along the DV boundary, where N and Wg are signaling)? How does membrane potential influence Smo subcellular localization?

We are pleased that the Reviewer finds our work novel and interesting. We have addressed many of the issues raised in the review pertaining to technical aspects of the data. As mentioned previously, we have eliminated the portions of the manuscript that deal with compartment-boundary maintenance and clone survival and focused it on the relationship between membrane potential and Hh signaling.

Specific comments

Fig. 1 Some images in this figure appear "blurred" (e.g A,A'D,E..) making it difficult to judge the data. Are these confocal images?

Imaging live tissue can result in less crisp images than fixed tissue, but we believe that the blurriness you are referring to could have been due to poor compression of the figure files. We have made sure to include high-resolution files in this submission.

Fig. 1F,G The authors claim that addition of ouabain leads to increased DiBAC signal. Where is the control for comparison, Fig. 1A or B?

The proper control has been included (Fig. 1F).

Fig. 1I The authors use a second sensor for membrane potential, ArcLight. The spatial pattern of this sensor's signal looks different to DiBAC. In particular, I do not see this one dorsoventral stripe of cells with higher signal intensity.

As we mentioned in the response to Reviewer 1, we have been concerned by the variability in ArcLight fluorescence and have removed those experiments from the revised manuscript.

Fig. 2E The authors claim that Rpk knockdown (limited to dorsal cells) diminishes DiBAC. However, I do not see a difference in DiBAC between dorsal and ventral cells. I also do not see evidence for an altered disc morphology (as the authors claim).

Please see response to Reviewer 1. We have removed this experiment and shown that knockdown using *rn-Gal4* or *dpp-Gal4* (using *Gal80ts*) does reduce DiBAC fluorescence.

Fig. 3B,C A control is missing: hh-AC/+ heterozygotes using the same temperature regime.

This control has been added (Fig. EV2).

Fig. 4B',B' Rpk-RNAi expression appears to only reduce Smo (B') or Ci (B') levels in the medial, but not lateral region of the dorsal pouch (also not in the hinge or notum)? Any explanation?

We believe this is consistent with the expression levels of the *ap-Gal4* driver in different regions of the disc (please see Fig. EV4A)

Also, I am a bit puzzled about the Ci staining in the control ventral region. The authors seem to use an Ci antibody recognizing the full-length Ci protein. However, the full-length Ci protein is present approximately only in the Ptc expressing cells, and not throughout the anterior compartment, as seems to be the case here. Any explanation?

We refer the reviewer to the original paper from the Kornberg lab (Aza-Blanc et al., 1997) that describes the staining pattern with this antibody (Please see Figure 1D in that paper). The full length protein is synthesized in all cells of the anterior compartment and is processed rapidly in most cells to the shorter form. It is stabilized in cells just anterior to the compartment boundary but detected at lower levels in all cells of the anterior compartment.

Fig. 4C The authors claim that Ptc expression was reduced in Rpk-RNAi cells, but that is not obvious. The authors should quantify fluorescence levels and use additional markers for Hedgehog signal transduction activity (dpp, engrailed). Is there a difference between 'high' and 'low'-level Hh-target genes?

Although the tissue is affected, we believe the difference in Ptc staining is clear. To further solidify this conclusion, we have also included anti-Ptc staining of a disc in which Rpk has been

knocked down in the wing pouch to further demonstrate the reduction in Ptc protein following Rpk knockdown (Fig. EV4 Supplement E-F').

Fig. 4D The authors predict that Irk-knockdown depolarizes cells. The authors should test this prediction by estimating membrane potential (DiBAC, ArcLight).

This data using Irk1 knockdown have been removed.

Fig. 4 B-D What is the evidence that these RNAi lines specifically targets Rpk/Irk1? Materials and Methods list two different Rpk-RNAi lines. Which was used in these experiments? What was the result with the other line? For Irk1, only one RNAi line is listed. Can the authors exclude cross-silencing by this line?

We have removed the data for Irk1 knockdown. For *rpk*, we show validity of the knockdown in Figure EV1.

Fig. 6 Is Rpk's function in maintaining a straight compartment boundary because its influence on Hedgehog signal transduction? Would re-establishment of Hedgehog signal transduction in Rpk knock-down cells (e.g. by co-expression of Ci-3m) restore normal boundary shape?

We have refocused our manuscript on the relationship between Hh signaling and membrane potential. We have removed the section on compartment-boundary maintenance.

Line 335 The authors suggest that "the zone of depolarization could be broader at earlier stages of development (Fig. 1k-K)". However, increased DiBAC fluorescence seems to be limited also at this early stage to the Ptc expressing, Hedgehog transducing cells (even though this is a bit unclear, see comment above). Is DiBAC fluorescence different between posterior cells and non-Ptc-expressing anterior cells (during early or late stages)?

DiBAC fluorescence appears similar between posterior cells and non-Ptc expressing anterior cells. However, DiBAC fluorescence does not allow us to detect subtle differences.

Fig. 7B A control is missing: What is the clone:compartment area ratio for control clones expressing only GFP (in particular for 96h)?

We have removed the entire section on differential survival of clones in the two compartments.

Fig. 7 The authors' results indicate that, when depolarized cells survive better in the anterior compartment, whereas, when hyperpolarized cells survive better in the posterior compartment. Again, is the degree of depolarization of anterior cells (excluding the Ptc expressing cells) different from that of posterior cells?

We have removed the entire section on differential survival of clones in the two compartments.

Reviewer #2 (Minor Comments):

Introduction: References are missing (e.g. line 52/53 "It has been suggested.." or lines 62/63 "Hh has a short range..")

These references have been added.

Reviewer #3 (General assessment and major comments (Required)):

Normal appendage development in Drosophila depends on the early segregation of A and P compartments, with P cells programmed to send Hh, which induces abutting A cells to produce morphogens such as Dpp. Hh is transduced by its receptor, Ptc, a multi-pass transporter-like protein that regulates the activity and cell-surface accumulation of the GPCR Smo by a still unknown mechanism. Sharp disparities in Hh signaling across the A/P boundary also play a role in preventing the cells from mixing, also by an unknown mechanism.

The present manuscript reports an intriguing observation that might help understand both mechanisms. Specifically, the authors have discovered that A cells that abut the A/P compartment boundary of the Drosophila wing are locally depolarized, correlating with their receipt of maximum Hh and their immiscibility with abutting P cells. Further, they present evidence that this local depolarization is induced by Hh signaling and plays a role in both Hh transduction and the segregation of A from P cells.

Unfortunately, the data supporting these conclusions are preliminary at best. Several key experiments are missing, the data for some of the experiments that are presented are problematic, and there is little if any evidence for a role of depolarization in maintaining the A/P boundary. As a consequence, I cannot recommend acceptance in eLife. The issues that would need to be resolved for the paper to merit publication are as follows.

1) Does Hh-dependent depolarization of A cells play a role in Hh signal transduction

The authors provide evidence that Hh signaling acts through its transcriptional effector Ci to induce A cells to up-regulate expression of the ENaC channel Rpk, correlating with their local depolarization. Specifically, they show that clones of cells that express a constitutively active form of the Ci are associated with local Rpk expression and possibly depolarization. Conversely, conditional reduction in Hh signaling has the opposite effect, causing a loss of Rpk up-regulation and depolarization. Although some of this data is problematic (see section 5, below),

the results are expected given the stereotypic Hh response pattern of both up-regulation of Rpk and depolarization. These results are consistent with, but don't provide evidence that, depolarization is involved in transduction.

*The author's argument for a role of Hh-induced Rpk-dependent membrane depolarization in transduction comes from the findings that RNAi knock down of Rpk suppresses this depolarization and appears to reduce Hh signaling outputs, whereas increasing membrane polarization by ectopic expression of *Irk1*, a Ki^+ channel enhances Hh signaling outputs. They also show corresponding effects of reducing or increasing *Vmem* on *Smo* accumulation and *Ptc* expression in salivary glands. They conclude that membrane depolarization caused by Hh-dependent up-regulation of Rpk is part of an amplification mechanism that enhances the response of A cells to Hh signal.*

A key expectation of this interpretation is that elevating expression of Rpk to levels corresponding to those normally observed in A cells at the A/P boundary should suffice to enhance the response to Hh signaling. Conversely, blocking the normal upregulation of Rpk activity at the boundary should reduce but not eliminate transduction, but have no consequence for A cells at a distance. These experiments are critical to confirming this interpretation and are technically feasible, but are lacking or inadequately performed.

*In particular, gain and loss of up-regulation experiments should be done in genetically marked clones, using physiologically relevant levels of Rpk and outputs that provide a quantitative assessment of Hh pathway activation (e.g., assessment of differences in expression of the classic *dpp-lacZ* enhancer trap insertion, which shows a long-range graded response to Hh signaling). This is because the surrounding w.t. tissue is needed to provide an internal control for quantitative differences, as well as an indication of the spatial parameters of the response.*

*It should be possible to generate marked *Flp* excision clones that ectopically express Rpk at its normal level of up-regulation at the A/P boundary. Similarly, loss of function clones could be generated using a knock-out allele of Rpk (easy to generate, using *Cas9/CRISPR* mutagenesis), and this could be done in the presence of a transgene that provides uniform low levels of Rpk corresponding to the levels seen away from the A/P boundary to, in effect, block the normal Hh-dependent up-regulation. Doing such experiments will require generating new transgenes and mutant alleles, but such experiments are necessary to assess the role of Rpk and depolarization in Hh transduction, both in the wing, and more generally in other appendages and segments. Note that making clones of *Gal4* driven Rpk might generate levels of Rpk far above the normal physiological levels, which can be assessed by antibody staining. If that is so, such a quick and dirty approach would fail to provide the necessary information.*

We thank the reviewer for this thoughtful review and these excellent suggestions for experiments. Unfortunately, the generation of the reagents necessary for many of these

experiments will take some time, especially since we are still working under reduced-density occupancy. As explained in the Introduction to the response, we have narrowed the scope of our work and mostly addressed technical issue raised in the reviews. The recent publication of the Spann et al paper has also necessitated the resubmission of our work with some urgency.

2) *What is the role of Rpk away from the A/P boundary.*

A related question is what, if anything, Rpk activity is doing in A cells away from the boundary, or in P cells, both of which seem to have similarly high membrane polarity. Given the possibility of other redundant channels (e.g., Ppk29), Rpk null clones might have little if any effect on Vmem away from the A/P boundary. On the other hand, the authors find that RNAi knock-down clones of Rpk appear compromised for viability throughout the A compartment despite appearing fine in P. This raises a number of questions that need to be resolved, as both A and P portions of the D compartment appears intact in apGal4 RNAi knock-down discs, and A cells along the A/P boundary also survive in dppGal4 RNAi knock-down discs (but see section 5). If, as the authors argue, clonal loss of Rpk expression in cells in any location within the A compartment causes them to become competitive "losers", but has no effect on P compartment cells, that complicates interpretation, as membrane polarity is normally high in both locations. Moreover, given the already high level of polarization of these cells, the low level of uninduced Rpk expression, and the likelihood of redundant channels, it's not clear whether such loss of function clone would cause membrane depolarization (something the authors do not assess). Is it possible that it membrane polarization becomes even higher in Rpk knock-down A but not in Rpk knock-down P cells? And if not, why would only the A cells become losers? More importantly, how does this relate to Hh transduction or compartment boundary stability, as these cells normally don't see Hh and are not near the boundary? If these Rpk RNAi knock down results are due to off target effects, or to other roles for Rpk, that presents further problems for interpretation.

We have removed the data that address differential clone survival in the two compartments; we agree that this work is still preliminary. However, we do note that Rpk appears to be expressed at lower levels in all cells and that knockdown of *rpk* using *ap-Gal4* increases Smo levels in both the anterior and posterior compartment.

3) *Is Hh-dependent depolarization of A cells a general mechanism for modulating Hh transduction and maintaining A/P boundaries?*

Another inadequately explored, but key, question is whether Hh-dependent Rpk induced depolarization is generally required for Hh transduction and compartment boundary maintenance, e.g., in other appendages. The authors were unable to obtain evidence for a role outside of the wing, possibly as they suggest because the DiBAC and ArcLight assays are not sufficiently sensitive. Analysis of loss and gain of function clones in the adult would provide a more stringent test, as adult patterning markers are exceptionally sensitive to altered Hh

transduction and loss of compartment boundary integrity. Such an analysis belongs in the paper. This is so, even if null or RNAi knock-down clones behave as competitive losers in A, whereas ectopic Rpk expressing clones behave as competitive losers in P. One would at least like to know if this is a general A vs P property. And also, to determine if it can be negated by any of several methods of generally reducing or eliminating competition between loser and winner cells (e.g., by performing the experiments in a Df(3L)H99/+ or Minute/+ backgrounds), and if so what the consequences are for patterning and boundary stability.

We have removed the section on compartment-boundary maintenance and differential clone survival in the revised manuscript.

4) Does Hh-dependent depolarization of A cells play a role in maintaining the compartment boundary

This is a key assertion of the paper, but is not well supported. Putting aside the different effects of gain or loss of depolarizing activity on cell competition, which in my view is a separate and unresolved matter, the main evidence comes down to a single finding. This is that the boundary appears disturbed in dppGal4 Rpk RNAi discs (Fig. 6B).

This is problematic for several reasons.

First, it is well established that the A/P boundary is maintained by two separate and redundant mechanisms: an abrupt disparity in Hh transduction and an inherent preference for A and P cells to avoid intermixing programmed by the selective activity of En. Either, alone, is sufficient to maintain a stable boundary, as documented by the behavior of smo- A compartment clones along the A/P boundary, which form smooth borders with A and P cells on either side, as well as by ptc- A clones away from the boundary. As a consequence, the disruption of the A/P boundary by dppGal4 Rpk RNAi knock-down cannot be attributed simply to a failure of the Hh signaling mechanism, as that should not suffice to destabilize the boundary.

Second, other experiments, in which Rpk RNAi knock-down is driven by apGal4, do not seem to disrupt the boundary (Fig. 4). This creates a problem for interpretation, as it seems to contradict the result of the dppGal4 knock-down. This discrepancy could be resolved, e.g., using ciGal4 and hhGal4 knock down, both alone and in combination to determine if it is the disparity as well as the direction of the disparity in knock-down activity that is responsible for the destabilization of the boundary.

Third, the apoptotic appearance of the GPF stain in dppGal4 Rpk RNAi discs raises the possibility that A cells at the A/P boundary, and possibly throughout the dppGal4 stripe, are dying. Minimally, this should be assessed, as there is clearly something going on with Rpk RNAi knock down and cell death, as revealed by the clonal experiments.

In sum, there is a clear expectation that selective loss of the Hh-based mechanism of compartmental segregation should cause A/P boundary phenotypes similar to the loss of Smo. The results do not show this. It is true that the authors have evidence for reciprocal effects of depolarizing or hyperpolarizing A versus P compartment clones for cell competition. But these are presently uninterpretable with respect to maintenance of the boundary because one of the most striking and still unexplained aspects of cell competition is that it is restricted to cells within a compartment: A cells do not compete with P cells.

We have removed the section on compartment-boundary maintenance and differential clone survival in the revised manuscript.

5) Weaknesses in the data

Figure 1

1A,B vs 1F,G. Comparing these images, it seems the general depolarization caused by ouabain is restricted to the wing pouch. Why should that be the case if DiBAC is a general indicator of depolarization?

The main point we are making is that ouabain abolishes the patterning of membrane potential in the wing disc. We are hesitant to draw strong conclusions when comparing the fluorescence intensity between two different preparations.

1H,I. The ap>ArcLight data is not convincing, as the fixed images also shows a diminished stripe in the vicinity of the A/P boundary. Also, both images should be counterstained with a marker for the A/P boundary (e.g., a hh-nuclear RFP or hh-mCherry knock in allele (both of which are published and available). Also, why not express the ArcLight under the control of ubiquitously expressed Gal4 driver as that would provide more information about what is happening in the rest of the disc, as well as in other discs. Perhaps even better would be to express ArcLight directly, on the control of a weak promoter. As it is, the DiBAC dye is far from ideal, so any way to improve visualizing Vmem would help.

We have found the variability in ArcLight fluorescence to be problematic and removed those data from the revised manuscript.

1K. The DiBAC stain of the early disc was not convincing, at least in terms of the width of the depolarized stripe, which is supposed to be broader than in the late discs.

In general, we find that the width of the DiBAC stripe matches the expression of high levels of Ptc. The use of live preparations for DiBAC makes accurate quantitation of width problematic.

Fig. 2

2E is not convincing. This is a key experiment, but depolarization along the A/P boundary is not visible in the V compartment. This experiment also needs a counterstain, e.g. a hh-nuclear-RFP or a hh-mCherry knock-in allele, as above.

This image has been removed. Instead we now show knockdown experiments using *dpp-Gal4* (Figure 2D) and *rn-Gal4* (Figure EV1), both done using Gal80ts to limit the adverse effects of long-term knockdown.

Fig. 3

3A is disconcerting in showing up-regulation of Rpk, but not ATP α , along the A/P boundary in the haltere.

The Na-K ATPase is typically observed on basolateral surfaces of epithelial cells and EnaC channels are often on the apical membrane. Thus ATP-alpha expression may simply have not been captured in the haltere disc at that plane of imaging. We have focused on the wing disc since these effects are most obvious in that tissue.

*3D,E is a perplexing result. The clones appear to be excluded from the wing blade, the region in which almost all of the other experiments are performed. The clones also appear to round up wherever they appear, and in some cases to cause elevated Rpk stain in cells surrounding the clone. For example, in E there is a second small clone surrounded by a circle of up-regulated ATP α . Another problem is that the confocal slices for the different stains appear to be different, as the upregulation around this second clone is not apparent in E'. This is also a problem in other figures. Yet another problem is that the upregulation of Rpk and ATP α along the A/P boundary is virtually undetectable. Finally, it looks like the Rpk stain has a high background that is elevated around folds, which undermines whether the accumulation in circles round the clones is real. At least for the Rpk stains, an *apGal4* RNAi knock-down experiment done in parallel should be shown to validate the Rpk antibody stain.*

We have now included validation of the Rpk antibody stain (Fig EV1). The confocal slice is the same for all images presented, and the upregulation of Rpk and ATPalpha at the compartment boundary is clearly visible but more subtle because the slice taken was optimized to show clones, and thus was a little more basal than the wing pouch. The Rpk antibody does show some background, but does not seem to be dramatically enriched in folds to the extent that it would preclude interpretation of clone staining.

3F,G. In this case the clones are not excluded from the wing, but the correspondence between the clones and the up-regulation of DiBAC stain is haphazard, at least when comparing the DiBAC only image, which looks, as in 3E' like a different focal plane.

In the revised manuscript we show images at lower clone density of clones expressing *ci3m* or *ptc-RNAi*. We show that with *ci3m*, we see increased DiBAC fluorescence in both compartments and with *ptc-RNAi* we see increased fluorescence only in the anterior compartment. Obtaining flat images with live preparations (needed for DiBAC) is challenging. We have provided quantification of comparison of fluorescence in the *ci3m* expressing clone and a neighboring region of the same area.

In general, I do not understand the choice of using UAS-Ci3m. Far better would have been simple ptc- clones (or less ideal, but likely still OK, ptcRNAi clones), as the level of activity of the Hh transduction pathway would have been the physiologically relevant maximum, unlike Ci3m, which could well be out of the physiological range. Interpretation of ptc- clones would also not be complicated by whatever ectopic Ci3m does in the P compartment.

We have included *ptc-RNAi* clones (Fig 3 and Fig EV3). These clones show increased DiBAC staining in the anterior compartment, and no change in staining in the posterior compartment as would be expected since *ptc* is not expressed in the posterior compartment.

Fig. 4

4B. apGal4 RNAi knock-down of Rpk reduces Smo accumulation in some, but not all of the D compartment. Inexplicably, neither Smo nor Ci appear to be reduced in D cells close to the D/V boundary (which should be counterstained, e.g., with Wg or Cut), and more surprisingly not in the dorsal hinge. Also disconcerting is the relatively uniform high accumulation of Smo and Ci ventrally, throughout the A compartment (in contrast with 4A). Also perplexing is the clear up-regulation of Ci in the hinge and notum, which is unexpected. Finally, the A/P boundary, as revealed by Ci in 4B and Ptc in 4C, appears smooth, complicating matters (see point 3 above). The same is true for 4D.

The *ap-Gal4* driver we used in this set of experiments does not drive in the second fold of the hinge, or in the notum. We have included an image of an *ap>GFP* disc, showing the expression domain (Fig. EV4A). We have also included discs in which *rpk* has been knocked down in the entire wing pouch using *rn-Gal4 Gal80ts*, showing clear reduction of Ci and Smo (Fig. EV4). We have removed the section on compartment boundary maintenance from the manuscript.

Here and in Figures 6 and 7 there should be some assessment of the adult phenotype, both in the wing and in other appendages (e.g., in the text or supplemental figures).

Knockdown of *rpk* with *ap-Gal4* is pupal-lethal, perhaps due to expression of the RNAi in the brain or other tissues.

Figs. 5 and S5

Ideally, these experiments, or at least a subset of them, should be done providing the blue light to only a portion of the gland in the image, perhaps shown at lower magnification, which should be possible. That would provide a necessary internal control, as salivary cell staining is sometimes erratic.

We have attempted these experiments. We have found that exposing just a portion of the gland results in increased Smo abundance in the entire gland, possibly because the cells are electrically coupled via gap junctions. We believe that the revised manuscript has been greatly strengthened by inclusion of optogenetic experiments in the wing disc (Figure 6).

Fig. 6. The Ci stain should be shown separately for both 6A and 6B, as it looks like it may be stabilized dorsally in 6B. Ptc expression should also be assayed, because reduced Hh transduction might result in increased spread of Hh in the A compartment, which could explain the up-regulation of Ci, which looks like it extends quite far. This also fits with what looks like an abnormal broadening of the dpp>GFPnls stripe. Finally, the quality of the high magnification images are inexcusably poor and the discs should be counterstained with DAPI, to make it possible to visualize picnotic nuclei in the abutting portions of both A and P. The same genotypes should also be assayed with a marker of apoptosis (e.g., DCP1).

We have removed the portion of the manuscript that deals with compartment-boundary maintenance.

Fig. 7.

7A,B Competition is associated with local cell death at the boundaries of loser or winner clones. Is this evident for NaChBac clones in 48hr or 72 hr clones, before they are lost? Is it suppressed by a Df(3R)H99/+ or Minute/+ background? Also the large clones in A" that survive in the A compartment appear to be excluded from the stripe of elevated Ci. Is this generally true, and if so, why should that be the case, as these clones should increase depolarization, and hence match the normal depolarization resulting from Rpk up-regulation. There is also the question of whether the NaChBac clones elevate Hh transduction. From the images shown, the clones that survive in or next to the region of elevated Ci accumulation do not appear to do so, though this would be easier to assess in Ci-only image, or even better, using the dpp-lacZ enhancer trap reporter. The authors should also confirm that these clones depolarize cells, as monitored by DiBAC.

7C,D. The A/P boundary appears to be stably maintained even though membrane potential should be diminished on both sides. Assuming reduced Vmem is confirmed by DiBAC staining, it suggests that a disparity in Vmem across the A/P boundary is not necessary to maintain the boundary

7F. Why are rpk-RNAi knock down clones excluded from the A compartment away from the compartment boundary, but not the P compartment, as Vmem is already high in these regions. Here too, is this competition associated with cell death in surviving clones in 48 and 72 hr discs, and is it eliminated in a Minute/+ or Df(3R)H99/+ background?

We agree that the work on differential clonal survival is preliminary and have removed it from the manuscript.

Reviewer #4 (General assessment and major comments (Required)):

The manuscript by Emmon-Bell et al. addresses the effect of membrane potential on cell fate and tissue patterning during development. The authors show patterned distribution of depolarized epithelial cells and find that two components that affect cell polarity interact with Hedgehog pathway signaling. They also employ various manipulations to demonstrate that depolarization promotes robust segregation of anterior and posterior cells in the Drosophila wing imaginal disc (but not other imaginal discs).

The claim that Hh response alters cell depolarization is based on observations with dyes and indicator proteins that are sensitive to polarity. Although the images are not always totally convincing, this claim seems reasonable, at least in the wing disc, as expression of several channels that affect polarity appear to be regulated by Hh signaling, as the authors astutely note from the data of Willsey et al. (eLife 2016). The authors provide no real explanation as to why similar Hh-dependent differences in polarity are not present in other imaginal tissues, leaving open the general question of how important or interesting this relationship is.

A second major claim, that the degree of cell depolarization alters Hh response, seems somewhat less well supported and again, this relationship is not universal (for example, cell depolarization differences along the D/V axis of the wing disc are not associated with differences in Hh response), and this raises the question of how important cell polarity truly is in modulating Hh response. Interesting speculations are offered (e.g., that polarity differences may affect the ion gradient-dependent activity of Ptc), but this begs the question of why Patched activity is not affected in some locations with clear differences in depolarization. The clearest demonstration of cell depolarization affecting Hh response is in the salivary gland using light-activated channels (optogenetics). Although this is not physiologic, the phenomena seem more clear, and this might have been an opportunity for the authors to more systematically explore the mechanism of the cell polarity effect on Hh response.

In a third major section of the paper, the authors present data suggesting that lineage segregation at the A/P compartment boundary is affected by cell polarity, and that cell polarity affects cell survival differentially to either side of the A/P compartment boundary. The authors do not resolve, however, whether these phenomena are due to effects on Hh response or some

other unrelated effect of cell polarity. Such an experiment might, for example, involve activating pathway by deleting or knocking down PKA in the same clones in which rpk is knocked down, to determine whether activation of Hh response rescues clone viability.

In sum, although some of these observations are quite intriguing, it is difficult to take away from the paper a clear notion of what is going on. If the insights could be pushed to a level of somewhat deeper understanding, particularly with regard to the effects of cell polarity on Hh response or on A/P compartment preference for clone survival, this might be worthy of eLife. Short of that deeper understanding, this manuscript remains somewhat preliminary.

We have removed the sections of the manuscript that deal with compartment boundary maintenance and differential clone survival – we agree that the work is a little preliminary. We have solidified the portion of the manuscript that deals with the relationship between membrane potential and Hedgehog signaling.

Below are a few specific comments:

1. Data in 3D,E, F and G indicated that DiBAC staining and expression of rpk and ATPa increase in response to a constitutively activated form of Ci. Is there a similar effect in the posterior compartment?

In the revised manuscript, we have generated clones that express either an activated form of Ci (*ci3m*) or *ptcRNAi*. With *ci3m* expression, we observe increased DiBAC fluorescence in both compartments. With *ptcRNAi*, we observe increased DiBAC fluorescence in anterior clones consistent with *ptc* being expressed only in anterior cells. (Figure 3 and Figure EV3)

2. Figure 3F' showed some variation in the degree of increased DiBAC staining among different RFP-marked clones. Some quantification of the effect should be provided to support the claim that high levels of Hh pathway activity reliably promote membrane depolarization.

A quantification of mean DiBAC fluorescence in Ci3M clones as compared to non-clone tissue has been added (Fig. EV3).

3. The data in Figure 5 suggest that changes of membrane potential can modulate Hh pathway activity in salivary gland cells. Although the change of Smo localization seems clear, expression of Ptc protein was only tested in the context of long-term expression of NaChBac and depolarization. Does Ptc expression also increase with short-term depolarization, as in panels M,N,O,P? The kinetics of the effect could provide insight into how directly Hh response is affected by changes in membrane potential.

In the revised manuscript we present optogenetic experiments in both the salivary gland and the wing disc (Figure 5, Figure EV5 and Figure 6). In both tissues we observe increased Smo

membrane association in both short-term and long-term experiments. We also show that this occurs independently of *ptc* since it is observed in the posterior compartment of the wing disc. We observe a robust increase in Ptc protein levels only in long-term experiments.

4. Typo in line 434, Boi.

This typo has been fixed (Line 385).

Dear Iswar,

Thank you once more for the submission of your revised manuscript to EMBO reports. It was evaluated again by former referee 1 and 2 and we have now received their reports, which are copied below.

As you will see, both referees acknowledge that the revised version has been much improved and support publication after a minor experimental and textual revision.

From the editorial side, there are also a few things that we need before we can proceed with the official acceptance of your study.

- Please provide up to five keywords.
- Please provide a conflict of interest statement/paragraph.
- Please provide a section on Author contributions.
- In that context, I noticed that one of the authors on the original submission was removed from the manuscript. Could you please indicate the reason for this change in authorship?
- Generally, the order of manuscript sections is currently incorrect. Please see our author guidelines for further information (<https://www.embopress.org/page/journal/14693178/authorguide#textformat>).
- Please reformat the references according to our Harvard style. See <https://www.embopress.org/page/journal/14693178/authorguide#referencesformat> for further information and for the respective EndNote file.
- There is a callout to Figure 8 in the text, which does not exist (line 372). Please correct this.
- Please remove the figure legends from the figure files.
- I attach to this email a related manuscript file with comments by our data editors. Please address all comments and upload a revised file with tracked changes with your final manuscript submission.
- Finally, EMBO reports papers are accompanied online by A) a short (1-2 sentences) summary of the findings and their significance, B) 2-3 bullet points highlighting key results and C) a synopsis image that is 550x200-600 pixels large (width x height) in .png format. You can either show a model or key data in the synopsis image. Please note that the size is rather small and that text needs to be readable at the final size. Please send us this information along with the revised manuscript.

With kind regards,

Martina

Referee #1:

The revised version of the manuscript is cleaner, more focused and the evidences presented support the conclusions. The authors have clarified the link between membrane potential (Vmem) and Hedgehog signaling after eliminating from the first version of the manuscript the most preliminary data, such as the role of depolarization in maintaining the edge of the A/P compartment border and the selective survival of clones with altered Vmem in each compartment.

The authors have performed most of the requested experiments to improve their work, including the proper controls and better images. They have also added new experimental data showing that the optogenetic manipulation of the membrane potential can modulate Hedgehog signaling both in the wing disc and in the salivary glands, demonstrating that Vmem regulates the location of the Smo in the membrane independently of Patched. These data are important to understand the mechanism by which Vmem regulates Hh signaling.

This work is indeed a step forward in understanding the mechanisms that regulate Hedgehog signaling during development. However, before being published in EMBO Report I have a few small requests that I hope will improve the manuscript:

Regarding experiments:

First, to explore the effect of Vmem in Wingless pathway, the authors have analyzed the expression of Wg and Cut, which are more closely related to the Notch pathway. The analysis of the expression of more specific Wg targets such as Senseless or Distaless could have given more reliable results.

Second, the authors have not looked at the effect of the ATP γ C knock down on Hh signaling. That is an obvious thing to do. Had the authors done so, they could have given an explanation of the results obtained, even if they were negative.

Regarding the text:

In the introduction section, the role of Hh signaling in the maintenance of the A/P compartment border is described. This description could be eliminated because the authors have removed from the manuscript the implication of the Vmem in the formation of the compartment border.

Referee #2:

The authors have satisfactorily addressed the comments.
However, the manuscript would benefit from the following revisions:

Abstract: The abstract emphasizes compartments and boundaries in the Drosophila wing disc. While the original manuscript version also addressed the mechanisms by which boundaries form, the revised version no longer does. The authors should consider revising the abstract to focus on Hedgehog signal transduction instead.

Introduction: Again, also the introduction emphasizes boundaries (first sentence: "During the development of many organisms, boundaries...."). The authors are encouraged to revise the introduction to introduce cell-cell signaling and the Hedgehog signal transduction pathway.

Discussion: Line 345, which refers to the segregation of A and P cells, should be deleted.

Minor comment:

Title should read "wing imaginal disc" (no hyphen).

Response to editorial requests

Dear Martina,

Thank you so much for all of your efforts in handling our manuscript. We are pleased that the reviewers found that our revised manuscript satisfied most of their concerns.

We have addressed all editorial requests and the remaining reviewer concerns.

I also wanted to address your question of why Riku Yasutomi is no longer an author. Riku did some of the experiments that were in the original submission to eLife (and then transferred to EMBO Reports) that pertained to the differential survival of depolarized and hyperpolarized clones in the two compartments. This whole section was removed based on reviewer comments since they were considered preliminary. Riku, therefore did not make any contribution to the manuscript in its current form. Maya, Riku and I discussed the situation and agreed to remove him as an author on this manuscript. He could build upon that work (which is very interesting) and publish it separately in the future.

Best wishes,

Iswar

Response to reviews

Referee #1:

The revised version of the manuscript is cleaner, more focused and the evidences presented support the conclusions. The authors have clarified the link between membrane potential (Vmem) and Hedgehog signaling after eliminating from the first version of the manuscript the most preliminary data, such as the role of depolarization in maintaining the edge of the A/P compartment border and the selective survival of clones with altered Vmem in each compartment.

The authors have performed most of the requested experiments to improve their work, including the proper controls and better images. They have also added new experimental data showing that the optogenetic manipulation of the membrane potential can modulate Hedgehog signaling both in the wing disc and in the salivary glands, demonstrating that Vmem regulates the location of the Smo in the membrane independently of Patched. These data are important to understand the mechanism by which Vmem regulates Hh signaling.

This work is indeed a step forward in understanding the mechanisms that regulate Hedgehog signaling during development. However, before being published in EMBO Report I have a few small requests that I hope will improve the manuscript:

Regarding experiments:

First, to explore the effect of Vmem in Wingless pathway, the authors have analyzed the expression of Wg and Cut, which are more closely related to the Notch pathway. The analysis of the expression of more specific Wg targets such as Senseless or Distaless could have given more reliable results.

The point we were trying to make was that the effects of membrane depolarization was not confined to the Hedgehog pathway but could also impact other signaling pathways. We did not want to convey the impression that we had explored the effect of depolarization on the Wingless pathway. The alterations in Wingless and Cut could not be explained by changes in Hedgehog signaling alone. In the revised manuscript we have clarified the text to indicate that the effect on Cut could be changes to Wingless or Notch.

Second, the authors have not looked at the effect of the ATP α knock down on Hh signaling. That is an obvious thing to do. Had the authors done so, they could have given an explanation of the results obtained, even if they were negative.

We have included these data in the revised manuscript and find that knocking down ATP α also disrupts Hedgehog signaling, but also severely disrupts the morphology of the disc. ATP α is a component of the Na/K-ATPase. As we had already shown in Figure 1, antagonizing the function of the Na/K-ATPase with ouabain abolishes patterning of

membrane potential throughout the disc and likely affects the physiology of all cells. Therefore this experiment is difficult to interpret.

Regarding the text:

In the introduction section, the role of Hh signaling in the maintenance of the A/P compartment border is described. This description could be eliminated because the authors have removed from the manuscript the implication of the Vmem in the formation of the compartment border.

Thank you – we have rewritten portions of the abstract and introduction to better reflect the focus of the revised manuscript.

Referee #2:

The authors have satisfactorily addressed the comments.

However, the manuscript would benefit from the following revisions:

Abstract: The abstract emphasizes compartments and boundaries in the Drosophila wing disc. While the original manuscript version also addressed the mechanisms by which boundaries form, the revised version no longer does. The authors should consider revising the abstract to focus on Hedgehog signal transduction instead.

Introduction: Again, also the introduction emphasizes boundaries (first sentence: "During the development of many organisms, boundaries...."). The authors are encouraged to revise the introduction to introduce cell-cell signaling and the Hedgehog signal transduction pathway.

Thank you – we have rewritten portions of the abstract and introduction to better reflect the focus of the revised manuscript.

Discussion: Line 345, which refers to the segregation of A and P cells, should be deleted.

This sentence has been deleted.

Minor comment:

Title should read "wing imaginal disc" (no hyphen).

We have made this change.

Iswar Hariharan
Univ. of California, Berkeley
Molecular and Cell Biology
515 LSA
Berkeley, CA 94720-3200
United States

Dear Iswar,

I am very pleased to accept your manuscript for publication in the next available issue of EMBO reports. Thank you for your contribution to our journal.

At the end of this email I include important information about how to proceed. Please ensure that you take the time to read the information and complete and return the necessary forms to allow us to publish your manuscript as quickly as possible.

As part of the EMBO publication's Transparent Editorial Process, EMBO reports publishes online a Review Process File to accompany accepted manuscripts. As you are aware, this File will be published in conjunction with your paper and will include the referee reports, your point-by-point response and all pertinent correspondence relating to the manuscript.

If you do NOT want this File to be published, please inform the editorial office within 2 days, if you have not done so already, otherwise the File will be published by default [contact: emboreports@embo.org]. If you do opt out, the Review Process File link will point to the following statement: "No Review Process File is available with this article, as the authors have chosen not to make the review process public in this case."

Should you be planning a Press Release on your article, please get in contact with emboreports@wiley.com as early as possible, in order to coordinate publication and release dates.

Thank you again for your contribution to EMBO reports and congratulations on a successful publication. Please consider us again in the future for your most exciting work.

Kind regards,
Martina

THINGS TO DO NOW:

You will receive proofs by e-mail approximately 2-3 weeks after all relevant files have been sent to

our Production Office; you should return your corrections within 2 days of receiving the proofs.

Please inform us if there is likely to be any difficulty in reaching you at the above address at that time. Failure to meet our deadlines may result in a delay of publication, or publication without your corrections.

All further communications concerning your paper should quote reference number EMBOR-2020-51861V3 and be addressed to emboreports@wiley.com.

Should you be planning a Press Release on your article, please get in contact with emboreports@wiley.com as early as possible, in order to coordinate publication and release dates.

Corresponding Author Name: Iswar K. Hariharan

Manuscript Number: 51861V1